# ROBUST CONDITIONAL GENERATIVE ADVERSARIAL NETWORKS

**Grigorios G. Chrysos**[1], **Jean Kossaifi**[1], **Stefanos Zafeiriou**[1]
[1] Department of Computing, Imperial College London, UK
`{g.chrysos, jean.kossaifi, s.zafeiriou}@imperial.ac.uk`

## ABSTRACT

Conditional generative adversarial networks (cGAN) have led to large improvements in the task of conditional image generation, which lies at the heart of computer vision. The major focus so far has been on performance improvement, while there has been little effort in making cGAN more robust to noise. The regression (of the generator) might lead to arbitrarily large errors in the output, which makes cGAN unreliable for real-world applications. In this work, we introduce a novel conditional GAN model, called *RoCGAN*, which leverages structure in the target space of the model to address the issue. Our model augments the generator with an unsupervised pathway, which promotes the outputs of the generator to span the target manifold even in the presence of intense noise. We prove that RoCGAN share similar theoretical properties as GAN and experimentally verify that our model outperforms existing state-of-the-art cGAN architectures by a large margin in a variety of domains including images from natural scenes and faces.

## 1 INTRODUCTION

Image-to-image translation and more generally conditional image generation lie at the heart of computer vision. Conditional Generative Adversarial Networks (cGAN) (Mirza & Osindero, 2014) have become a dominant approach in the field, e.g. in dense[1] regression (Isola et al., 2017; Pathak et al., 2016; Ledig et al., 2017; Bousmalis et al., 2016; Liu et al., 2017; Miyato & Koyama, 2018; Yu et al., 2018; Tulyakov et al., 2018). They accept a source signal as input, e.g. prior information in the form of an image or text, and map it to the target signal (image). The mapping of cGAN does not constrain the output to the target manifold, thus the output can be arbitrarily off the target manifold (Vidal et al., 2017). This is a critical problem both for academic and commercial applications. To utilize cGAN or similar methods as a production technology, we need to study their generalization even in the face of intense noise.

Similarly to regression, classification also suffers from sensitivity to noise and lack of output constraints. One notable line of research consists in complementing supervision with unsupervised learning modules. The unsupervised module forms a new pathway that is trained with the same, or different data samples. The unsupervised pathway enables the network to explore the structure that is not present in the labelled training set, while implicitly constraining the output. The addition of the unsupervised module is only required during the training stage and results in no additional computational cost during inference. Rasmus et al. (2015) and Zhang et al. (2016) modified the original bottom-up (encoder) network to include top-down (decoder) modules during training. However, in dense regression both bottom-up and top-down modules exist by default, and such methods are thus not trivial to extend to regression tasks.

Motivated by the combination of supervised and unsupervised pathways, we propose a novel conditional GAN which includes implicit constraints in the latent subspaces. We coin this new model '*Robust Conditional GAN*' (*RoCGAN*). In the original cGAN the generator accepts a source signal and maps it to the target domain. In our work, we (implicitly) constrain the decoder to generate samples that span only the target manifold. We replace the original generator, i.e. encoder-decoder,

---

[1]The output includes at least as many dimensions as the input, e.g. super-resolution, or text-to-image translation.

with a two pathway module (see Fig. 1). The first pathway, similarly to the cGAN generator, performs regression while the second is an autoencoder in the target domain (unsupervised pathway). The two pathways share a similar network structure, i.e. each one includes an encoder-decoder network. The weights of the two decoders are shared which promotes the latent representations of the two pathways to be semantically similar. Intuitively, this can be thought of as constraining the output of our dense regression to span the target subspace. The unsupervised pathway enables the utilization of all the samples in the target domain even in the absence of a corresponding input sample. During inference, the unsupervised pathway is no longer required, therefore the testing complexity remains the same as in cGAN.

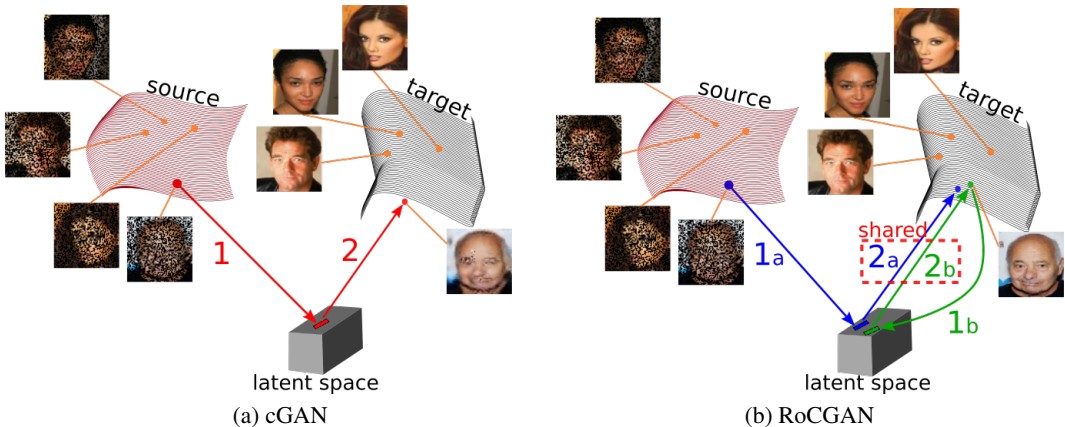

(a) cGAN                     (b) RoCGAN

Figure 1: The mapping process of the generator of the baseline cGAN (in (a)) and our model (in (b)). (a) The source signal is embedded into a low-dimensional, latent subspace, which is then mapped to the target subspace. The lack of constraints might result in outcomes that are arbitrarily off the target manifold. (b) On the other hand, in RoCGAN, steps 1b and 2b learn an autoencoder in the target manifold and by sharing the weights of the decoder, we restrict the output of the regression (step 2a). All figures in this work are best viewed in color.

In the following sections, we introduce our novel RoCGAN and study their (theoretical) properties. We prove that RoCGAN share similar theoretical properties with the original GAN, i.e. convergence and optimal discriminator. An experiment with synthetic data is designed to visualize the target subspaces and assess our intuition. We experimentally scrutinize the sensitivity of the hyper-parameters and evaluate our model in the face of intense noise. Moreover, thorough experimentation with both images from natural scenes and human faces is conducted in two different tasks. We compare our model with both the state-of-the-art cGAN and the recent method of Rick Chang et al. (2017). The experimental results demonstrate that RoCGAN outperform the baseline by a large margin in all cases.

Our contributions can be summarized as following:

- We introduce RoCGAN that leverages structure in the target space. The goal is to promote robustness in dense regression tasks.

- We scrutinize the model performance under (extreme) noise and adversarial perturbations. To the authors' knowledge, this robustness analysis has not been studied previously for dense regression.

- We conduct a thorough experimental analysis for two different tasks. We outline how RoCGAN can be used in a semi-supervised learning task, how it performs with lateral connections from encoder to decoder.

**Notation**: Given a set of $N$ samples, $s^{(n)}$ denotes the $n^{th}$ conditional label, e.g. a prior image; $y^{(n)}$ denotes the respective target image. Unless explicitly mentioned otherwise $|| \cdot ||$ will declare an $\ell_1$ norm. The symbols $\mathcal{L}_*$ define loss terms, while $\lambda_*$ denote regularization hyper-parameters optimized on the validation set.

## 2 RELATED WORK

Conditional image generation is a popular task in computer vision, dominated by approaches similar to cGAN. Apart from cGAN, the method by Isola et al. (2017), widely known as 'pix2pix', is the main alternative. Pix2pix includes three modifications over the baseline cGAN: i) lateral skip connections between the encoder and the decoder network are added in the generator, ii) the discriminator accepts pairs of source/gt and source/model output images, iii) additional content loss terms are added. The authors demonstrate how those performance related modifications can lead to an improved visual outcome. Despite the improved performance, the problem with the additional guarantees remains the same. That is we do not have any direct supervision in the process, since both the latent subspace and the projection are learned; the only supervision is provided by the ground-truth (gt) signal in the generator's output.

Adding regularization terms in the loss function can impose stronger supervision, thus restricting the output. The most frequent regularization term is feature matching, e.g. perceptual loss (Ledig et al., 2017; Johnson et al., 2016), or embeddings for faces (Schroff et al., 2015). Feature matching minimizes the distance between the projection of generated and ground-truth signals. However, the pre-defined feature space is restrictive. The method introduced by Salimans et al. (2016) performs feature matching in the discriminator; the motivation lies in matching the low-dimensional distributions created by the discriminator layers. Matching the discriminator's features has demonstrated empirical success. However, this does not affect the generator and its latent subspaces directly.

A new line of research that correlates with our goals is that of adversarial attacks (Szegedy et al., 2014; Yuan et al., 2017; Samangouei et al., 2018). It is observed that perturbing input samples with a small amount of noise, often imperceptible to the human eye, can lead to severe classification errors. There are several techniques to 'defend' against such attacks. A recent example is the Fortified networks of Lamb et al. (2018) which uses Denoising Autoencoders (Vincent et al., 2008) to ensure that the input samples do not fall off the target manifold. Kumar et al. (2017) estimate the tangent space to the target manifold and use that to insert invariances to the discriminator for classification purposes. Even though RoCGAN share similarities with those methods, the scope is different since a) the output of our method is high-dimensional[2] and b) adversarial examples are not extended to dense regression[3].

Except for the study of adversarial attacks, combining supervised and unsupervised learning has been used for enhancing the classification performance. In the Ladder network, Rasmus et al. (2015) modify a typical bottom-up network for classification by adding a decoder and lateral connections between the encoder and the decoder. During training they utilize the augmented network as two pathways: i) labelled input samples are fed to the initial bottom-up module, ii) input samples are corrupted with noise and fed to the encoder-decoder with the lateral connections. The latter pathway is an autoencoder; the idea is that it can strengthen the resilience of the network to samples outside the input manifold, while it improves the classification performance.

Our core goal consists in constraining the model's output. Aside from deep learning approaches, such constraints in manifolds were typically tackled with component analysis. Canonical correlation analysis (Hotelling, 1936) has been extensively used for finding common subspaces that maximally correlate the data (Panagakis et al., 2016). The recent work of Murdock et al. (2018) combines the expressiveness of neural networks with the theoretical guarantees of classic component analysis.

## 3 METHOD

In this section, we elucidate our proposed RoCGAN. To make the paper self-contained we first review the original conditional GAN model (sec. 3.1), before introducing RoCGAN (sec. 3.2). Sequentially, we pose the modifications required in case of shortcut connections from the encoder to the decoder (sec. 3.3). In sec. 3.4 we assess the intuition behind our model with synthetic data. The core idea in RoCGAN is to leverage structure in the output space of the model. We achieve that by

---

[2]In the classification tasks studied, e.g. the popular Imagenet (Deng et al., 2009), there are up to a thousand classes, while our output includes tens or hundreds of thousands of dimensions.

[3]The robustness in our case refers to being resilient to changes in the distribution of the labels (label shift) and training set (covariance shift) (Wang et al., 2017).

replacing the single pathway in the generator with two pathways. In the appendix, we study the theoretical properties of our method and prove that RoCGAN share the same properties as the original GAN (Goodfellow et al., 2014).

## 3.1 CONDITIONAL GAN

GAN consist of a generator and a discriminator module commonly optimized with alternating gradient descent methods. The generator samples $\boldsymbol{z}$ from a prior distribution $p_{\boldsymbol{z}}$, e.g. uniform, and tries to model the target distribution $p_d$; the discriminator $D$ tries to distinguish between the samples generated from the model and the target (ground-truth) distributions. Conditional GAN (cGAN) (Mirza & Osindero, 2014) extend the formulation by providing the generator with additional labels. In cGAN the generator $G$ typically takes the form of an encoder-decoder network, where the encoder projects the label into a low-dimensional latent subspace and the decoder performs the opposite mapping, i.e. from low-dimensional to high-dimensional subspace. If we denote $\boldsymbol{s}$ the conditioning label and $\boldsymbol{y}$ a sample from the target distribution, the adversarial loss is expressed as:

$$
\begin{aligned}
\mathcal{L}_{adv}(\boldsymbol{G}, \boldsymbol{D}) = \mathbb{E}_{\boldsymbol{s}, \boldsymbol{y} \sim p_d(\boldsymbol{s}, \boldsymbol{y})}[\log \boldsymbol{D}(\boldsymbol{y}|\boldsymbol{s})] + \\
\mathbb{E}_{\boldsymbol{s} \sim p_d(\boldsymbol{s}), \boldsymbol{z} \sim p_z(\boldsymbol{z})}[\log(1 - \boldsymbol{D}(\boldsymbol{G}(\boldsymbol{s}, \boldsymbol{z})|\boldsymbol{s}))]
\end{aligned}
\tag{1}
$$

by solving the following min-max problem:

$$
\begin{aligned}
\min_{\boldsymbol{w}_G} \max_{\boldsymbol{w}_D} \mathcal{L}_{adv}(\boldsymbol{G}, \boldsymbol{D}) = \min_{\boldsymbol{w}_G} \max_{\boldsymbol{w}_D} \mathbb{E}_{\boldsymbol{s}, \boldsymbol{y} \sim p_d(\boldsymbol{s}, \boldsymbol{y})}[\log \boldsymbol{D}(\boldsymbol{y}|\boldsymbol{s}, \boldsymbol{w}_D)] + \\
\mathbb{E}_{\boldsymbol{s} \sim p_d(\boldsymbol{s}), \boldsymbol{z} \sim p_z(\boldsymbol{z})}[\log(1 - \boldsymbol{D}(\boldsymbol{G}(\boldsymbol{s}, \boldsymbol{z}|\boldsymbol{w}_G)|\boldsymbol{s}, \boldsymbol{w}_D))]
\end{aligned}
$$

where $\boldsymbol{w}_G, \boldsymbol{w}_D$ denote the generator's and the discriminator's parameters respectively. To simplify the notation, we drop the dependencies on the parameters and the noise $\boldsymbol{z}$ in the rest of the paper.

The works of Salimans et al. (2016) and Isola et al. (2017) demonstrate that auxiliary loss terms, i.e. feature matching and content loss, improve the final outcome, hence we consider those as part of the vanilla cGAN. The feature matching loss (Salimans et al., 2016) is:

$$
\mathcal{L}_f = \sum_{n=1}^{N} ||\pi(\boldsymbol{G}(\boldsymbol{s}^{(n)})) - \pi(\boldsymbol{y}^{(n)})||
\tag{2}
$$

where $\pi()$ extracts the features from the penultimate layer of the discriminator.

The final loss function for the cGAN is the following:

$$
\mathcal{L} = \mathcal{L}_{adv} + \lambda_c \cdot \underbrace{\sum_{n=1}^{N} ||\boldsymbol{G}(\boldsymbol{s}^{(n)}) - \boldsymbol{y}^{(n)}||}_{content-loss} + \lambda_\pi \cdot \mathcal{L}_f
\tag{3}
$$

where $\lambda_c, \lambda_\pi$ are hyper-parameters to balance the loss terms.

## 3.2 ROCGAN

Just like cGAN, RoCGAN consist of a generator and a discriminator. The generator of RoCGAN includes two pathways instead of the single pathway of the original cGAN. The first pathway, referred as *reg pathway* henceforth, performs a similar regression as its counterpart in cGAN; it accepts a sample from the source domain and maps it to the target domain. We introduce an additional unsupervised pathway, named *AE pathway*. AE pathway works as an autoencoder in the target domain. Both pathways consist of similar encoder-decoder networks[4]. By sharing the weights of their decoders, we promote the regression outputs to span the target manifold and not induce arbitrarily

---

[4]In principle the encoders' architectures might differ, e.g. when the two domains differ in dimensionality; however, in our case they share the same architectures.

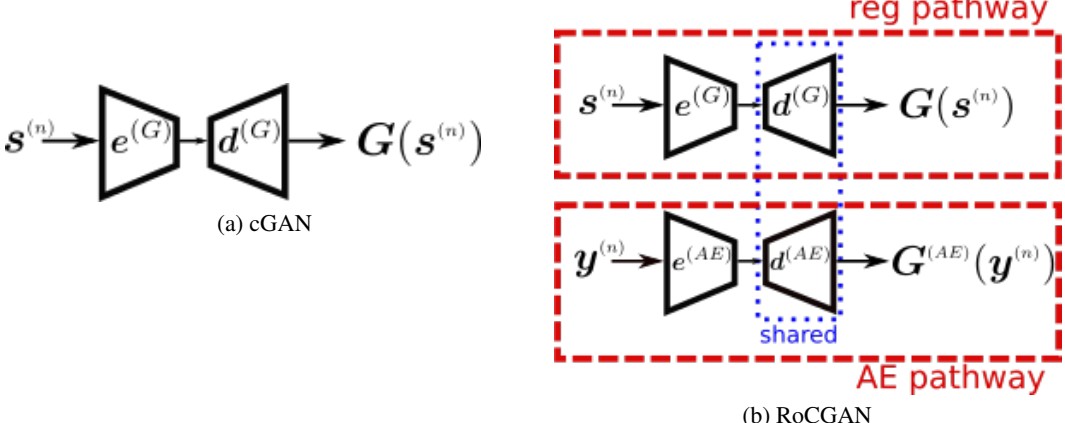

Figure 2: Schematic of the generator of (a) cGAN versus (b) our proposed RoCGAN. The single pathway of the original model is replaced with two pathways.

large errors. A schematic of the generator is illustrated in Fig. 2. The discriminator can remain the same as the cGAN: it accepts the reg pathway's output along with the corresponding target sample as input.

To simplify the notation below, the superscript 'AE' abbreviates modules of the AE pathway and 'G' modules of the reg pathway. We denote $G(s^{(n)}) = d^{(G)}(e^{(G)}(s^{(n)}))$ the output of the reg pathway and $G^{(AE)}(y^{(n)}) = d^{(AE)}(e^{(AE)}(y^{(n)}))$ the output of the AE pathway.

The unsupervised module (autoencoder in the target domain) contributes the following loss term:

$$\mathcal{L}_{AE} = \sum_{n=1}^{N} [f_d(y^{(n)}, G^{(AE)}(y^{(n)}))] \tag{4}$$

where $f_d$ denotes a divergence metric (in this work an $\ell_1$ loss).

Despite sharing the weights of the decoders, we cannot ensure that the latent representations of the two pathways span the same space. To further reduce the distance of the two representations in the latent space, we introduce the latent loss term $\mathcal{L}_{lat}$. This term minimizes the distance between the encoders' outputs, i.e. the two representations are spatially close (in the subspace spanned by the encoders). The latent loss term is:

$$\mathcal{L}_{lat} = \sum_{n=1}^{N} ||e^{(G)}(s^{(n)}) - e^{(AE)}(y^{(n)})|| \tag{5}$$

The final loss function of RoCGAN combines the loss terms of the original cGAN (eq. 3) with the additional two terms for the AE pathway:

$$\mathcal{L}_{RoCGAN} = \mathcal{L}_{adv} + \lambda_c \cdot \underbrace{\sum_{n=1}^{N} ||G(s^{(n)}) - y^{(n)}||}_{content-loss} + \lambda_\pi \cdot \mathcal{L}_f + \lambda_{ae} \cdot \mathcal{L}_{AE} + \lambda_l \cdot \mathcal{L}_{lat} \tag{6}$$

As a future step we intend to replace the latent loss term $\mathcal{L}_{lat}$ with a kernel-based method (Gretton et al., 2007) or a learnable metric for matching the distributions (Ma et al., 2018).

### 3.3 ROCGAN WITH SKIP CONNECTIONS

The RoCGAN model of sec. 3.2 describes a family of networks and not a predefined set of layers. A special case of RoCGAN emerges when skip connections are included in the generator. In the next few paragraphs, we study the modification required, i.e. an additional loss term.

Skip connections are frequently used as they enable deeper layers to capture more abstract representations without the need of memorizing all the information. Nevertheless, the effects of the skip connections in the representation space have not been thoroughly studied. The lower-level representations are propagated directly to the decoder through the shortcut, which makes it harder to train the longer path (Rasmus et al., 2015), i.e. the network excluding the skip connections.

This challenge can be implicitly tackled by maximizing the variance captured by the longer path representations. To that end, we add a loss term that penalizes the correlations in the representations (of a layer) and thus implicitly encourage the representations to capture diverse and useful information. We implement the decov loss introduced by Cogswell et al. (2016):

$$\mathcal{L}_{decov} = \frac{1}{2}(||\boldsymbol{C}||_F^2 - ||diag(\boldsymbol{C})||_2^2) \tag{7}$$

where $diag()$ computes the diagonal elements of a matrix and $\boldsymbol{C}$ is the covariance matrix of the layer's representations. The loss is minimized when the covariance matrix is diagonal, i.e. it imposes a cost to minimize the covariance of hidden units without restricting the diagonal elements that include the variance of the hidden representations.

A similar loss is explored by Valpola (2015), where the decorrelation loss is applied in every layer. Their loss term has stronger constraints: i) it favors an identity covariance matrix but also ii) penalizes the smaller eigenvalues of the covariance more. We have not explored this alternative loss term, as the decov loss worked in our case without the additional assumptions of the Valpola (2015).

### 3.4 EXPERIMENT ON SYNTHETIC DATA

We design an experiment on synthetic data to explore the differences between the original generator and our novel two pathway generator. Specifically, we design a network where each encoder/decoder consists of two fully connected layers; each layer followed by a RELU. We optimize the generators only, to avoid adding extra learned parameters.

The inputs/outputs of this network span a low-dimensional space, which depends on two independent variables $x, y \in [-1, 1]$. We've experimented with several arbitrary functions in the input and output vectors and they perform in a similar way. We showcase here the case with input vector $[x, y, e^{2x}]$ and output vector $[x + 2y + 4, e^x + 1, x + y + 3, x + 2]$. The reg pathway accepts the three inputs, projects it into a two-dimensional space and the decoder maps it to the target four-dimensional space.

We train the baseline and the autoencoder modules separately and use their pre-trained weights to initialize the two pathway network. The loss function of the two pathway network consists of the $\mathcal{L}_{lat}$ (eq. 5) and $\ell_2$ content losses in the two pathways. The networks are trained either till convergence or till $100,000$ iterations (batch size $128$) are completed.

During testing, $6,400$ new points are sampled and the overlaid results are depicted in Fig. 3; the individual figures for each output can be found in the appendix. The $\ell_1$ errors for the two cases are: $9,843$ for the baseline and $1,520$ for the two pathway generator. We notice that the two pathway generator approximates the target manifold better with the same number of parameters during inference.

## 4 EXPERIMENTS

**Implementation details**: To provide a fair comparison to previous cGAN works, our implementation is largely based on the conventions of Isola et al. (2017); Salimans et al. (2016); Zhu et al. (2017). A 'layer' refers to a block of three units: a convolutional unit with a $4 \times 4$ kernel size, followed by Leaky RELU and batch normalization (Ioffe & Szegedy, 2015). To obtain RoCGAN, we augment a vanilla cGAN model as follows: i) we duplicate the encoder/decoder; ii) we share the decoder's weights in

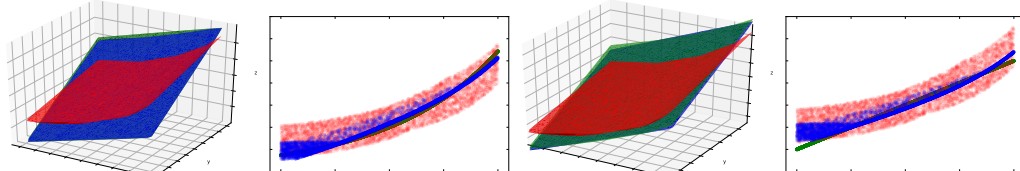

Figure 3: Qualitative results in the synthetic experiment of sec. 3.4. Each plot corresponds to the respective manifolds in the output vector; the first and third depend on both $x, y$ (xyz plot), while the rest on $x$ (xz plot). The green color visualizes the target manifold, the red the baseline and the blue ours. Even though the two models include the same parameters during inference, the baseline does not approximate the target manifold as well as our method.

the two pathways; iii) we add the additional loss terms. The values of the additional hyper-parameters are $\lambda_l = 25$, $\lambda_{ae} = 100$ and $\lambda_{decov} = 1$; the common hyper-parameters with the vanilla cGAN, e.g. $\lambda_c$, $\lambda_\pi$, remain the same. The decov loss is applied in the output of the encoder, which in our experimentation did minimize the correlations in the longer path. The rest hyper-parameters remain the same as in the baseline.

We conduct a number of auxiliary experiments in the appendix. Specifically, an ablation study on the significance and the sensitivity of the hyper-parameters is conducted; additional architectures are implemented, while we evaluate our model under more intense noise. In addition, we extend the concept of adversarial examples in regression and verify that our model is more resilient to them than the baseline. The results demonstrate that our model accepts a range of hyper-parameter values, while it is robust to additional sources of noise.

We experiment with two categories of images with significant applications: images from i) natural scenes and ii) faces. In the natural scenes case, we constrain the number of training images to few thousand since frequently that is the scale of the labelled examples available. The network used in the experiments below, dumped '4layer', consists of four layers in the decoder, while the decoder followed by four layers in the decoder.

Two inverse tasks, i.e. denoising and sparse inpainting, are selected for our quantitative evaluation. During training, the images are corrupted, for the two tasks, in the following way: for denoising $25\%$ of the pixels in each channel are uniformly dropped; for sparse inpainting $50\%$ of the pixels are converted to black. During testing, we evaluate the methods in two settings: i) similar corruption as they were trained, ii) more intense corruption, i.e. we drop $35\%$ of the pixels in the denoising case and $75\%$ of the pixels in the sparse inpainting case. The widely used image quality loss (SSIM) (Wang et al., 2004) is used as a quantitative metric. We train and test our method against the i) baseline cGAN, ii) the recent strong-performing OneNet (Rick Chang et al., 2017). OneNet uses an ADMM learned prior, i.e. it projects the corrupted prior images into the subspace of natural images to guide the ADMM solver. In addition, we train an Adversarial Autoencoder (AAE) (Makhzani et al., 2015) as an established method capable of learning compressed representations. Each module of the AAE shares the same architecture as its cGAN counterpart, while the AAE is trained with images in the target space. During testing, we provide the ground-truth images as input and use the reconstruction for the evaluation. In our experimental setting, AAE can be thought of as an upper performance limit of RoCGAN/cGAN for a given capacity (number of parameters).

## 4.1 NATURAL SCENES

We train the '4layer' baseline/RoCGAN with images from natural scenes, both indoors and outdoors. The $4,900$ samples of the VOC 2007 Challenge (Everingham et al., 2010) form the training set, while the $10,000$ samples of tiny ImageNet (Deng et al., 2009) consist the testing set.

The quantitative evaluation with SSIM is presented in Tab. 1. OneNet (Rick Chang et al., 2017) does not perform as well as the baseline or our model. From our experimentation this can be attributed to the projection to the manifold of natural images that is not trivial, however it is more resilient to additional noise than the baseline. In both inverse tasks RoCGAN improve the baseline cGAN results by a margin of $0.05$ ($10 - 13\%$ relative improvement). When we apply additional corruption in the testing images, RoCGAN are more robust with a considerable improvement over the baseline. This

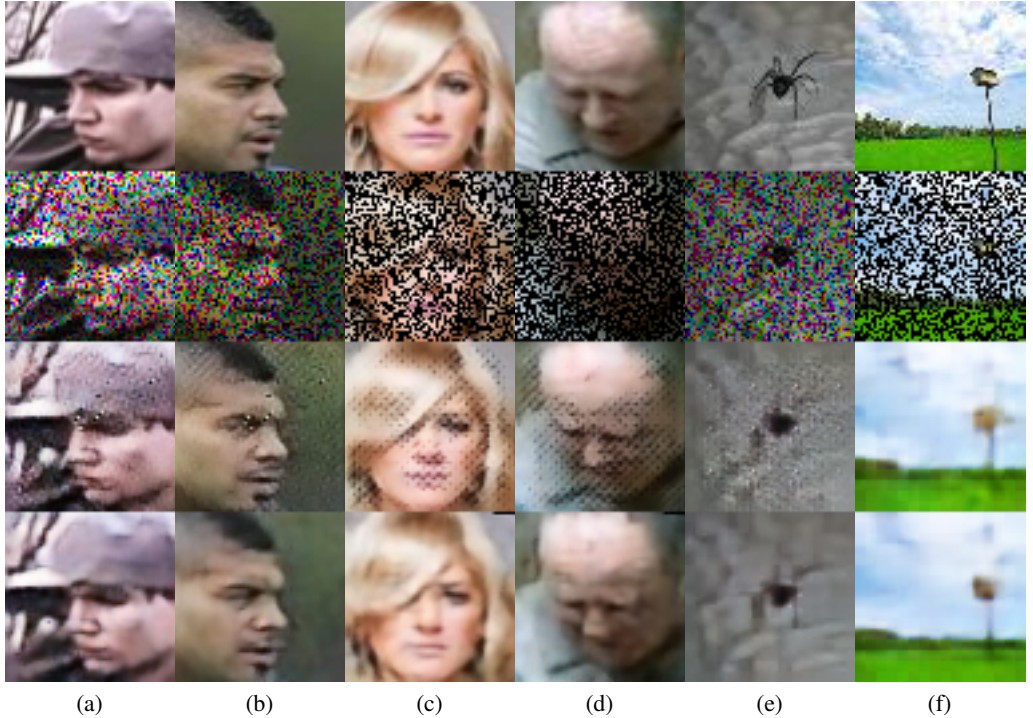

(a)        (b)        (c)        (d)        (e)        (f)

Figure 4: Qualitative results (*best viewed in color*). The first row depicts the target image, the second row the corrupted one (used as input to the methods). The third row depicts the output of the baseline cGAN, while the outcome of our method is illustrated in the fourth row. There are different evaluations visualized for faces: (a) denoising, (b) denoising with additional noise at test time, (c) sparse inpainting, (d) sparse inpainting with $75\%$ black pixels. For natural scenes the columns (e) and (f) denote the denoising and sparse inpainting results respectively.

can be attributed to the implicit constraints of the AE pathway, i.e. the decoder is more resilient to approximating the target manifold samples.

| Method     *Obj. / Task* | Faces | | | | Natural Scenes | | | |
|---|---|---|---|---|---|---|---|---|
| | Denoising | | Sparse Inpaint. | | Denoising | | Sparse Inpaint. | |
| | $25\%$ | $35\%$ | $50\%$ | $75\%$ | $25\%$ | $35\%$ | $50\%$ | $75\%$ |
| Rick Chang et al. (2017) | 0.758 | 0.748 | 0.701 | 0.682 | 0.591 | 0.574 | 0.585 | 0.535 |
| Baseline-4layer | 0.803 | 0.765 | 0.801 | 0.701 | 0.628 | 0.599 | 0.639 | 0.542 |
| Ours-4layer | 0.834 | 0.821 | 0.804 | 0.708 | 0.668 | 0.654 | 0.648 | 0.548 |
| AAE | 0.866 | | | | 0.702 | | | |

Table 1: Quantitative results in the '4layer' network in both faces and natural scenes cases. For both 'objects' we compute the SSIM. In both denoising and sparse inpainting, the leftmost evaluation is the one with corruptions similar to the training, while the one on the right consists of samples with additional corruptions, e.g. in denoising 35% of the pixels are dropped.

## 4.2 FACES

In this experiment we utilize the MS-Celeb (Guo et al., 2016) as the training set ($3, 4$ million samples), and the whole Celeb-A (Liu et al., 2015) as the testing set ($202, 500$ samples). The large datasets enable us to validate our model extensively in a wide range of faces.

We use the whole training set to train the two compared methods (Baseline-4layer and Ours-4layer) and the 4-layer AAE. The results of the quantitative evaluation exist in table 1. Our method outperforms both the baseline and OneNet by a significant margin; the difference increases when evaluated with more intense corruptions. The reason that the sparse inpainting task appears to have a smaller improvement remains elusive; in the different architectures in the appendix our model has similar performance in the two tasks. We include the AAE as an upper limit of the representation capacity of the architecture. The AAE result specifies that with the given architecture the performance can be up to 0.866.

## 5 CONCLUSION

We introduce the Robust Conditional GAN (RoCGAN) model, a new conditional GAN capable of leveraging unsupervised data to learn better latent representations, even in the face of large amount of noise. RoCGAN's generator is composed of two pathways. The first pathway (*reg pathway*), performs the regression from the source to the target domain. The new, added pathway (*AE pathway*) is an autoencoder in the target domain. By adding weight sharing between the two decoders, we implicitly constrain the reg pathway to output images that span the target manifold. The linear analogy along with the synthetic experiment demonstrate how RoCGAN can create more robust results, while we prove that our model shares similar convergence properties with generative adversarial networks. The ablation study dictates that our model is more resilient to intense noise and more robust to adversarial examples than the baseline. The experimental results with images (natural scenes and faces) showcase that RoCGAN outperform existing, state-of-the-art conditional GAN models.

## 6 ACKNOWLEDGEMENTS

We would like to thank Markos Georgopoulos for our fruitful conversations during the preparation of this work. The work of Grigorios Chrysos was partially funded by an Imperial College DTA. The work of Stefanos Zafeiriou was partially funded by the EPSRC Fellowship DEFORM: Large Scale Shape Analysis of Deformable Models of Humans (EP/S010203/1) and a Google Faculty Award.

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

## A    INTRODUCTION

In this following sections (of the appendix) we include additional insights, a theoretical analysis along with additional experiments. The sections are organized as following:

- In sec. B we validate our intuition for the RoCGAN constraints through the linear equivalent.
- A theoretical analysis is provided in sec. C.
- We implement different networks in sec. D to assess whether the performance gain can be attributed to a single architecture.
- An ablation study is conducted in sec. E comparing the hyper-parameter sensitivity and the robustness in the face of extreme noise.

The Fig. 5, 6, 7, 8 include all the outputs of the synthetic experiment of the main paper. As a reminder, the output vector is $[x + 2y + 4, e^x + 1, x + y + 3, x + 2]$ with $x, y \in [-1, 1]$.

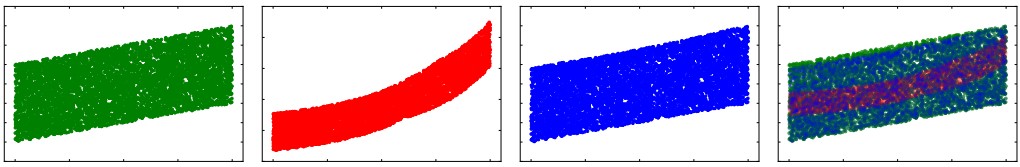

Figure 5: Qualitative results in the synthetic experiment (main paper). Output of the $1^{st}$ function. From left to right: The target (ground-truth) curve in green, the output of the single pathway network (baseline) in red, the two pathway network in blue and all three overlaid. The output vector of the $1^{st}$ and the $3^{rd}$ functions are plotted here with respect to $x$, the full 3D plot is in the manuscript. All figures in this work are best viewed in color.

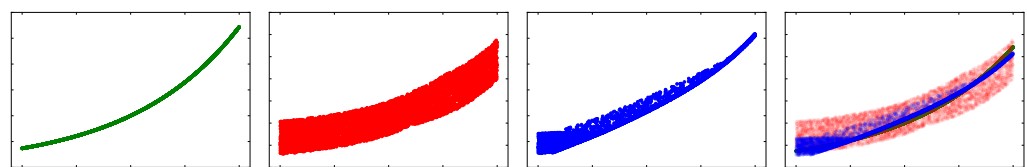

Figure 6: Qualitative results in the synthetic experiment (main paper). Output of the $2^{nd}$ function. See Fig. 5 for details.

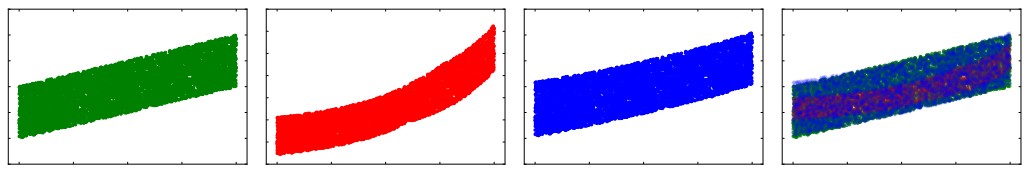

Figure 7: Qualitative results in the synthetic experiment (main paper). Output of the $3^{rd}$ function. See Fig. 5 for details.

## B    LINEAR GENERATOR ANALOGY

The exact nature and convergence properties of deep networks remain elusive (Vidal et al., 2017), however we can study the linear equivalent of deep methods to build on our intuition. To that end, we explore the linear equivalent of our method. Since the discriminator in RoCGAN can remain the same as in the baseline cGAN, we focus in the generators. To perform the analysis on the linear equivalent, we simply drop the piecewise non-linear units in the generators.

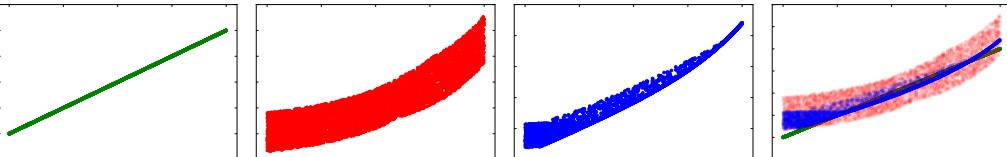

Figure 8: Qualitative results in the synthetic experiment (main paper). Output of the $4^{th}$ function. See Fig. 5 for details.

We assume a network with two encoding and two decoding layers in this section; all layers include only linear units. The symbols $\boldsymbol{W}_l^{(G)}$ with $l \in [1, 4]$ are the $l^{th}$ layer's parameters (reg pathway). The linear autoencoder (AE) has a similar structure; $\boldsymbol{W}_l^{(AE)}$ denote the respective parameters for the AE. We denote with $\boldsymbol{X}$ the input signal, with $\boldsymbol{Y}$ the target signal and $\hat{\boldsymbol{Y}}$ the AE output, $\tilde{\boldsymbol{Y}}$ the regression output. Then:

$$\hat{\boldsymbol{Y}} = \underbrace{\boldsymbol{W}_4^{(AE)} \boldsymbol{W}_3^{(AE)}}_{\boldsymbol{U}_D^T} \underbrace{\boldsymbol{W}_2^{(AE)} \boldsymbol{W}_1^{(AE)}}_{\boldsymbol{U}_E} \boldsymbol{Y} \tag{8}$$

is the reconstruction of the autoencoder and

$$\tilde{\boldsymbol{Y}} = \underbrace{\boldsymbol{W}_4^{(G)} \boldsymbol{W}_3^{(G)}}_{\boldsymbol{U}_{D,(G)}^T} \underbrace{\boldsymbol{W}_2^{(G)} \boldsymbol{W}_1^{(G)}}_{\boldsymbol{U}_{E,(G)}} \boldsymbol{X} \tag{9}$$

is the regression of the generator (reg pathway). We define the auxiliary $\boldsymbol{U}_{D,(G)}^T = \boldsymbol{W}_4^{(G)} \boldsymbol{W}_3^{(G)}$, $\boldsymbol{U}_{E,(G)} = \boldsymbol{W}_2^{(G)} \boldsymbol{W}_1^{(G)}$, $\boldsymbol{U}_D^T = \boldsymbol{W}_4^{(AE)} \boldsymbol{W}_3^{(AE)}$ and $\boldsymbol{U}_E = \boldsymbol{W}_2^{(AE)} \boldsymbol{W}_1^{(AE)}$. Then Eq. 8 and 9 can be written as:

$$\begin{cases} \tilde{\boldsymbol{Y}} = \boldsymbol{U}_{D,(G)}^T \boldsymbol{U}_{E,(G)} \boldsymbol{X} \\ \hat{\boldsymbol{Y}} = \boldsymbol{U}_D^T \boldsymbol{U}_E \boldsymbol{Y} \end{cases} \tag{10}$$

The AE approximates under mild condition robustly the target manifold of the data Bengio et al. (2013). If we now define $\boldsymbol{U}_{D,(G)} = \boldsymbol{U}_D$, then the output of the generator $\tilde{\boldsymbol{Y}}$ spans the subspace of $\boldsymbol{U}_D$.

Given that $\boldsymbol{U}_{D,(G)} = \boldsymbol{U}_D$, we constrain the output of the generator to lie in the subspaces learned with the AE.

To illustrate how a projection to a target subspace can contribute to constraining the image, the following visual example is designed. We learn a PCA model using one hundred thousand images from MS-Celeb; we do not apply any pose normalization or alignment (out of the paper's scope). We maintain 90% of the variance. In Fig. 9 we sample a random image from Celeb-A and downscale it[5]; we use bi-linear interpolation to upscale it to the original dimensions. We project and reconstruct both the original and the upscaled versions; note that the output images are similar. This similarity illustrates how the linear projection forces both images to span the same subspace.

## C    THEORETICAL ANALYSIS

In the next few paragraphs, we prove that RoCGAN share the properties of the original GAN (Goodfellow et al., 2014). We derive the optimal discriminator and then compute the optimal value of $\mathcal{L}_{adv}(\boldsymbol{G}, \boldsymbol{D})$.

---

[5]Downscaling is used as a simple corruption. Other corruptions, e.g. gaussian blurring, have similar results.

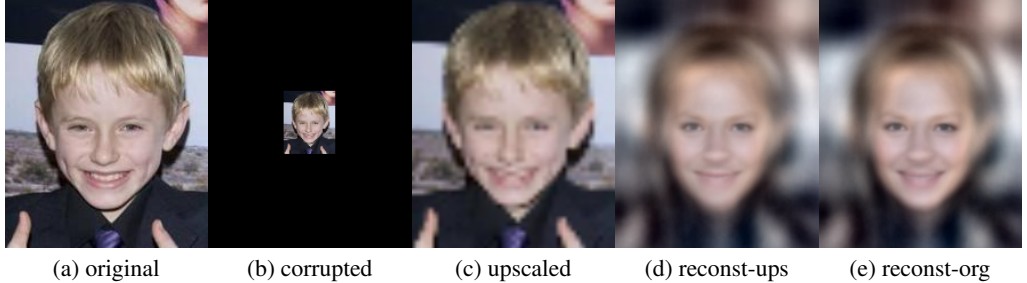

| (a) original | (b) corrupted | (c) upscaled | (d) reconst-ups | (e) reconst-org |

Figure 9: Projection to linear subspace. The original image on the left was corrupted (downscaled $\times 4$). The image was upscaled with bi-linear interpolation in (c). Both the original and the upscaled images are projected and reconstructed from a PCA. The reconstruction of the original image ('reconst-org') along with the linearly upscaled ('reconst-ups') image are similar. The simple linear projection demonstrates how constraining the output through a linear subspace can result in a more robust output.

**Proposition 1.** *If we fix the generator $\boldsymbol{G}$ (reg pathway), the optimal discriminator is:*

$$\boldsymbol{D}^* = \frac{p_d(\boldsymbol{s}, \boldsymbol{y})}{p_d(\boldsymbol{s}, \boldsymbol{y}) + p_g(\boldsymbol{s}, \boldsymbol{y})} \tag{11}$$

where $p_g$ is the model (generator) distribution.

*Proof.* Since the generator is fixed, the goal of the discriminator is to maximize the $\mathcal{L}_{adv}$ where:

$$\mathcal{L}_{adv}(\boldsymbol{G}, \boldsymbol{D}) = \int_{\boldsymbol{y}} \int_{\boldsymbol{s}} p_d(\boldsymbol{y}, \boldsymbol{s}) \log \boldsymbol{D}(\boldsymbol{y}|\boldsymbol{s}) d\boldsymbol{y} d\boldsymbol{s} + \int_{\boldsymbol{s}} \int_{\boldsymbol{z}} p_d(\boldsymbol{s}) p_z(\boldsymbol{z}) \log(1 - \boldsymbol{D}(\boldsymbol{G}(\boldsymbol{s}, \boldsymbol{z})|\boldsymbol{s})) d\boldsymbol{s} d\boldsymbol{z} =$$
$$\int_{\boldsymbol{y}} \int_{\boldsymbol{s}} p_d(\boldsymbol{s}, \boldsymbol{y}) \log \boldsymbol{D}(\boldsymbol{y}|\boldsymbol{s}) d\boldsymbol{y} + p_g(\boldsymbol{s}, \boldsymbol{y}) \log(1 - \boldsymbol{D}(\boldsymbol{y}|\boldsymbol{s})) d\boldsymbol{y} d\boldsymbol{s} \tag{12}$$

To maximize the $\mathcal{L}_{adv}$, we need to optimize the integrand above. We note that with respect to $\boldsymbol{D}$ the integrand has the form $f(y) = a \cdot log(y) + b \cdot log(1 - y)$. The function $f$ for $a, b \in (0, 1)$ as in our case, obtains a global maximum in $\frac{a}{a+b}$, so:

$$\mathcal{L}_{adv}(\boldsymbol{G}, \boldsymbol{D}) \leq \int_{\boldsymbol{y}} \int_{\boldsymbol{s}} p_d(\boldsymbol{s}, \boldsymbol{y}) \log \boldsymbol{D}^*(\boldsymbol{y}|\boldsymbol{s}) d\boldsymbol{y} + p_g(\boldsymbol{s}, \boldsymbol{y}) \log(1 - \boldsymbol{D}^*(\boldsymbol{y}|\boldsymbol{s})) d\boldsymbol{y} d\boldsymbol{s} \tag{13}$$

with

$$\boldsymbol{D}^* = \frac{p_d(\boldsymbol{s}, \boldsymbol{y})}{p_d(\boldsymbol{s}, \boldsymbol{y}) + p_g(\boldsymbol{s}, \boldsymbol{y})} \tag{14}$$

thus $\mathcal{L}_{adv}$ obtains the maximum with $\boldsymbol{D}^*$. $\qquad \square$

**Proposition 2.** *Given the optimal discriminator $\boldsymbol{D}^*$ the global minimum of $\mathcal{L}_{adv}$ is reached if and only if $p_g = p_d$, i.e. when the model (generator) distribution matches the data distribution.*

*Proof.* From proposition 1, we have found the optimal discriminator as $\boldsymbol{D}^*$, i.e. the $\arg\max_{\boldsymbol{D}} \mathcal{L}_{adv}$. If we replace the optimal value we obtain:

$$\max_{\boldsymbol{D}} \mathcal{L}_{adv}(\boldsymbol{G}, \boldsymbol{D}) = \int_{\boldsymbol{y}} \int_{\boldsymbol{s}} p_d(\boldsymbol{s}, \boldsymbol{y}) \log \boldsymbol{D}(\boldsymbol{y}|\boldsymbol{s}) d\boldsymbol{y} + p_g(\boldsymbol{s}, \boldsymbol{y}) \log(1 - \boldsymbol{D}(\boldsymbol{y}|\boldsymbol{s})) d\boldsymbol{y} d\boldsymbol{s} =$$

$$\int_{\boldsymbol{y}} \int_{\boldsymbol{s}} p_d(\boldsymbol{s}, \boldsymbol{y}) \log(\frac{p_d(\boldsymbol{s}, \boldsymbol{y})}{p_d(\boldsymbol{s}, \boldsymbol{y}) + p_g(\boldsymbol{s}, \boldsymbol{y})}) + p_g(\boldsymbol{s}, \boldsymbol{y}) \log(1 - \frac{p_d(\boldsymbol{s}, \boldsymbol{y})}{p_d(\boldsymbol{s}, \boldsymbol{y}) + p_g(\boldsymbol{s}, \boldsymbol{y})}) d\boldsymbol{y} d\boldsymbol{s} = \quad (15)$$

$$\int_{\boldsymbol{y}} \int_{\boldsymbol{s}} p_d(\boldsymbol{s}, \boldsymbol{y}) \log(\frac{p_d(\boldsymbol{s}, \boldsymbol{y})}{p_d(\boldsymbol{s}, \boldsymbol{y}) + p_g(\boldsymbol{s}, \boldsymbol{y})}) + p_g(\boldsymbol{s}, \boldsymbol{y}) \log(\frac{p_g(\boldsymbol{s}, \boldsymbol{y})}{p_d(\boldsymbol{s}, \boldsymbol{y}) + p_g(\boldsymbol{s}, \boldsymbol{y})}) d\boldsymbol{y} d\boldsymbol{s}$$

We add and subtract $\log(2)$ from both terms, which after few math operations provides:

$$\max_{\boldsymbol{D}} \mathcal{L}_{adv}(\boldsymbol{G}, \boldsymbol{D}) = - \int_{\boldsymbol{y}} \int_{\boldsymbol{s}} (p_d(\boldsymbol{s}, \boldsymbol{y}) + p_g(\boldsymbol{s}, \boldsymbol{y})) \log(2) d\boldsymbol{y} d\boldsymbol{s} +$$

$$\int_{\boldsymbol{y}} \int_{\boldsymbol{s}} (p_d(\boldsymbol{s}, \boldsymbol{y}) \log \frac{\frac{p_d(\boldsymbol{s}, \boldsymbol{y}) + p_g(\boldsymbol{s}, \boldsymbol{y})}{2}}{p_d(\boldsymbol{s}, \boldsymbol{y})} + p_g(\boldsymbol{s}, \boldsymbol{y}) \log \frac{\frac{p_d(\boldsymbol{s}, \boldsymbol{y}) + p_g(\boldsymbol{s}, \boldsymbol{y})}{2}}{p_g(\boldsymbol{s}, \boldsymbol{y})}) d\boldsymbol{y} d\boldsymbol{s} = \quad (16)$$

$$-2 \cdot \log(2) + KL(p_d || \frac{p_d + p_g}{2}) + KL(p_g || \frac{p_d + p_g}{2})$$

where in the last row KL symbolizes the Kullback–Leibler divergence. The latter one can be rewritten more conveniently with the help of the Jensen–Shannon (JSD) divergence as

$$\max_{\boldsymbol{D}} \mathcal{L}_{adv}(\boldsymbol{G}, \boldsymbol{D}) = -\log(4) + 2 \cdot JSD(p_d || p_g) \quad (17)$$

The Jensen–Shannon divergence is non-negative and obtains the zero value only if $p_d = p_g$. Equivalently, $\max_{\boldsymbol{D}} \mathcal{L}_{adv}(\boldsymbol{G}, \boldsymbol{D}) \geq -\log(4)$ and has a global minimum (under the constraint that the discriminator is optimal) when $p_d = p_g$.

$\square$

## D ADDITIONAL EXPERIMENTS

In this section, we describe additional experimental results and details.

In addition to the SSIM metric, we use the $\ell_1$ loss to measure the loss in the experiments of the main paper. The results in table 2 confirm that RoCGAN outperform both compared methods. The larger difference in the cases of more intense noise demonstrates that our model is indeed robust to additional cases not trained on. Additional visualizations are provided in Fig. 10.

| Obj. / Task
Method | Faces | | | | Natural Scenes | | | |
| --- | --- | --- | --- | --- | --- | --- | --- | --- |
| | Denoising | | Sparse Inpaint. | | Denoising | | Sparse Inpaint. | |
| | 25% | 35% | 50% | 75% | 25% | 35% | 50% | 75% |
| Rick Chang et al. (2017) | 701.6 | 801.6 | 838.3 | 876.1 | 1005.9 | 1083.4 | 1207.1 | 1401.1 |
| Baseline-4layer | 652.7 | 728.4 | 618.8 | 846.5 | 983.5 | 1057.7 | 949.4 | 1352.6 |
| Ours-4layer | 590.3 | 632.1 | 616.1 | 830.0 | 918.8 | 968.9 | 939.4 | 1351.9 |

Table 2: Quantitative results in the '4layer' network in both faces and natural scenes cases. In this table, the $\ell_1$ loss is reported. In each task, the leftmost evaluation is the one with corruptions similar to the training, while the one on the right consists of samples with additional corruptions, e.g. in denoising 35% of the pixels are dropped.

Unless otherwise mentioned, the experiments in the following paragraphs are conducted in the face case, while the evaluation metrics remain the same as in the main paper, i.e. the noise during training/testing and the SSIM evaluation metric.

| Task / Method | Denoising | | Sparse Inpaint. | |
|---|---|---|---|---|
| | 25% | 35% | 50% | 75% |
| Baseline-4layer | 0.657 | 0.643 | 0.629 | 0.514 |
| Ours-4layer | 0.679 | 0.666 | 0.645 | 0.534 |
| AAE | 0.700 | | | |

Table 3: Quantitative results for Imagenet (sec. D.1).

| Task / Method | Denoising | | Sparse Inpaint. | |
|---|---|---|---|---|
| | 25% | 35% | 50% | 75% |
| Baseline-5layer | 0.851 | 0.826 | 0.819 | 0.707 |
| Ours-5layer | 0.890 | 0.884 | 0.873 | 0.818 |
| Baseline-6layer | 0.859 | 0.843 | 0.816 | 0.727 |
| Ours-6layer | 0.881 | 0.865 | 0.882 | 0.822 |
| Baseline-4layer-skip | 0.885 | 0.863 | 0.855 | 0.726 |
| Ours-4layer-skip | 0.896 | 0.881 | 0.859 | 0.744 |

Table 4: Additional quantitative results (SSIM, see main paper) for the following protocols: i) '5layer' network, ii) 50 thousand training images, iii) skip connections.

## D.1 LARGE SCALE NATURAL IMAGES

To delineate further the performance of our model in different settings, we conduct an experiment with Imagenet (Russakovsky et al., 2015), a large dataset for natural images. We utilize the training set of Imagenet which consists of $1, 2$ million images and its testset that includes 98 thousand images.

The experimental results are depicted in table 3. The outcomes corroborate the experimental results of the main paper, as RoCGAN outperforms the cGAN in both tasks. We note that the AAE works as the upper limit of the methods and denotes the representation power that the given encoder-decoder can reach.

## D.2 DIFFERENT ARCHITECTURES

To assess whether RoCGAN's improvement is network-specific, we implement different architectures including more layers. The goal of this work is not to find the best performing architecture, thus we do not employ an exhaustive search in all proposed cGAN models. Our goal is to propose an alternative model to the baseline cGAN and evaluate how this works in different networks.

We implement three additional networks which we coin '5layer', '6layer' and '4layer-skip'. Those include five, six layers in the encoder/decoder respectively, while the '4layer-skip' includes a lateral connection from the output of the third encoding layer to the input of the second decoding layer. The first two increase the capacity of the network, while the '4layer-skip' implements the modification for the skip case in the '4layer' network[6].

We evaluate these three networks as in the '4layer' network (main paper); the results are added in table 4. Notice that both '5layer' and '6layer' networks improve their counterpart in the '4layer' case, however the '6layer' networks do not improve their '5layer' counterpart. This can be partly attributed to the increased difficulty of training deeper networks without additional regularization techniques (He et al., 2016). In addition, we emphasize that the denoising and the sparse inpainting results cannot be directly compared, since they correspond to different a) types and b) amount of corruption in all evaluations. Nevertheless, the improvement in the sparse inpainting with additional noise is impressive, given that the hyper-parameters are optimized for the denoising case (see sec E). The most critical observation is that in all cases our model consistently outperforms the baseline. The

---

[6]We test RoCGAN and the respective baseline for 2, 4, 5, 6 layer cases covering a range of deeper networks.

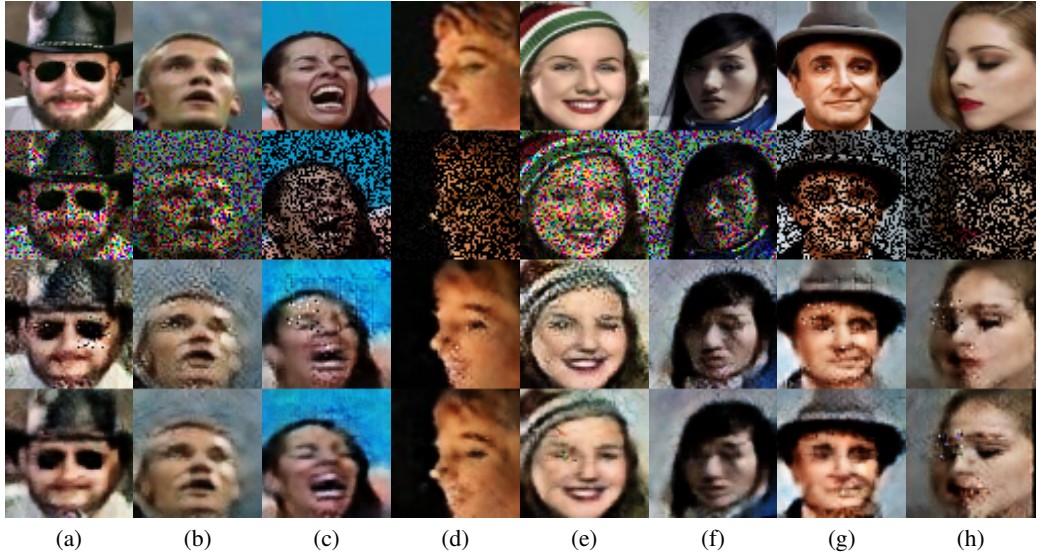

|     |     |     |     |     |     |     |     |
| --- | --- | --- | --- | --- | --- | --- | --- |
| (a) | (b) | (c) | (d) | (e) | (f) | (g) | (h) |

Figure 10: Qualitative results; best viewed in color. The first row depicts the ground-truth image, the second row the corrupted one (input to methods), the third the output of the baseline cGAN, the fourth illustrates the outcome of our method. The four first columns are based on the protocol of '4layer' network, while the four rightmost columns on the protocol '4layer-50k'. There are different evaluations visualized for faces: (a), (e) Denoising, (b), (f) denoising with augmented noise at test time, (c), (g) sparse inpainting, (d), (h) sparse inpainting with 75% black pixels.

difference is increasing under additional noise during inference time with up to **15**% performance improvement observed in the sparse inpainting case.

### D.3 SEMI-SUPERVISED TRAINING

A side-benefit of our new model is the ability to utilize unsupervised data to learn the AE pathway. Collecting unlabelled data in the target domain is frequently easier than finding pairs of corresponding samples in the two domains. To that end, we test whether RoCGAN support such semi-supervised learning.

We randomly pick $50,000$ labelled images while we use the rest three million as unlabelled. The 'label' in our case is the corrupted images. The baseline model is trained with the labelled $50,000$ samples. RoCGAN model is trained with $50,000$ images in the reg pathway, while the AE pathway with all the available (unlabelled) samples.

Table. 5 includes the quantitative results of the semi-supervised case. As expected the performance in most experiments drops from the full training case, however we observe that the performance in RoCGAN decreases significantly less than cGAN ('baseline-4layer-50k'). In other words, RoCGAN can benefit greatly from additional examples in the target domain. We hypothesize that this enables the AE pathway to learn a more accurate representation, which is reflected to the final RoCGAN outcome.

### D.4 ALTERNATIVE ERROR METRIC

In our experimental setting every input image should be mapped (close) to its target image. To assess the domain-specific performance for faces, we utilize the cosine distance distribution plot (CDDP).

One of the core features in images of faces is the identity of the person. We utilize the well-studied recognition embeddings (Schroff et al., 2015) to evaluate the similarity of the target image with the outputs of compared methods. The ground-truth identities in our case are not available in the embeddings' space; we consider instead the target image's embedding as the ground-truth for each

| Method \ Task | Denoising | | Sparse Inpaint. | |
|---|---|---|---|---|
| | 25% | 35% | 50% | 75% |
| Baseline-4layer-50k | 0.788 | 0.747 | 0.798 | 0.617 |
| Ours-4layer-50k | 0.829 | 0.813 | 0.813 | 0.681 |

Table 5: Quantitative results for the semi-supervised training of RoCGAN (sec. D.3). The difference of the two models is increased (in comparison to the fully supervised case). RoCGAN utilize the additional unsupervised data to improve the mapping between the domains even with less corresponding pairs.

Table 6: Details of the generator for the '4layer' network baseline. Our modified generator includes in the AE pathway the same parameters. The parameters mentioned below are valid also for the '4layer-50k' and the '4layer-skip' networks. 'Filter size' denotes the size of the convolutional filters; the last number denotes the number of output filters. BN stands for batch normalization. Conv denotes a convolutional layer, while F-Conv denotes a transposed convolutional layer with fractional-stride.

(a) Encoder

| Layer | Filter Size | Stride | BN |
|---|---|---|---|
| Conv. 1 | $4 \times 4 \times 64$ | 4 | $\times$ |
| Conv. 2 | $4 \times 4 \times 128$ | 2 | $\checkmark$ |
| Conv. 3 | $4 \times 4 \times 256$ | 2 | $\checkmark$ |
| Conv. 4 | $4 \times 4 \times 512$ | 4 | $\checkmark$ |

(b) Decoder

| Layer | Filter Size | Stride | BN |
|---|---|---|---|
| F-Conv. 1 | $1 \times 1 \times 256$ | 4 | $\checkmark$ |
| F-Conv. 2 | $4 \times 4 \times 128$ | 2 | $\checkmark$ |
| F-Conv. 3 | $4 \times 4 \times 64$ | 2 | $\checkmark$ |
| F-Conv. 4 | $4 \times 4 \times 3$ | 4 | $\times$ |

comparison. The plot is constructed as follows: For each pair of output and corresponding target image, we compute the cosine distance of their embeddings; the cumulative distribution of those distances is plotted. Mathematically the distance of the $n^{th}$ pair is formulated as:

$$\mathcal{F}(\boldsymbol{y}^{(n)}, \boldsymbol{o}^{(n)}) = \frac{< \boldsymbol{\Phi}(\boldsymbol{y}^{(n)}), \boldsymbol{\Phi}(\boldsymbol{o}^{(n)}) >}{||\boldsymbol{\Phi}(\boldsymbol{y}^{(n)})||_2 \cdot ||\boldsymbol{\Phi}(\boldsymbol{o}^{(n)})||_2} \qquad (18)$$

where $\boldsymbol{o}^{(n)}$ denotes the output of each method, $\boldsymbol{y}^{(n)}$ the respective target image and $\boldsymbol{\Phi}$ is the function computing the embedding. A perfect reconstruction per comparison, e.g. $\mathcal{F}(\boldsymbol{y}^{(n)}, \boldsymbol{y}^{(n)})$, would yield a plot of a Dirac delta around one; a narrow distribution centered at one denotes proximity to the target images' embeddings.

The plot with the CDDP is visualized in Fig. 11 for the '4layer' case as detailed in the main paper. The results illustrate that AAE has embeddings that are closer to the target embeddings as expected; from the compared methods the RoCGAN outperform the cGAN in the proximity to the target embeddings.

## D.5    ADDITIONAL EXPERIMENTAL DETAILS

All the images utilized in this work are resized to $64 \times 64 \times 3$. In the case of natural scenes, instead of rescaling the images during the training stage, we crop random patches in every iteration from the image. We utilize the ADAM optimizer with a learning rate of $2 \cdot 10^{-5}$ for all our experiments. The batch size is $128$ for images of faces and $64$ for the natural scenes.

In table 6 the details about the layer structure for the '4layer' generator are provided; the other networks include similar architecture as depicted in tables 7, 8. The discriminator retains the same structure in all the experiments in this work (see table 9).

## E    ABLATION STUDY

In the following paragraphs we conduct an ablation study to assess RoCGAN in different cases, i.e. effect of hyper-parameters, loss terms, additional noise. Unless mentioned otherwise, the architecture used is the '4layer' network. The experiments are in face denoising with the similarity metric (SSIM) and the setup similar to the main paper comparisons.

Table 7: Details of the generator for the '5layer' network baseline.

(a) Encoder

| Layer | Filter Size | Stride | BN |
|---|---|---|---|
| Conv. 1 | $4 \times 4 \times 32$ | 2 | $\times$ |
| Conv. 2 | $4 \times 4 \times 64$ | 2 | $\checkmark$ |
| Conv. 3 | $4 \times 4 \times 128$ | 2 | $\checkmark$ |
| Conv. 4 | $4 \times 4 \times 256$ | 2 | $\checkmark$ |
| Conv. 5 | $4 \times 4 \times 768$ | 4 | $\checkmark$ |

(b) Decoder

| Layer | Filter Size | Stride | BN |
|---|---|---|---|
| F-Conv. 1 | $1 \times 1 \times 256$ | 4 | $\checkmark$ |
| F-Conv. 2 | $4 \times 4 \times 128$ | 2 | $\checkmark$ |
| F-Conv. 3 | $4 \times 4 \times 64$ | 2 | $\checkmark$ |
| F-Conv. 4 | $4 \times 4 \times 32$ | 2 | $\checkmark$ |
| F-Conv. 5 | $4 \times 4 \times 3$ | 2 | $\times$ |

Table 8: Details of the generator for the '6layer' network baseline.

(a) Encoder

| Layer | Filter Size | Stride | BN |
|---|---|---|---|
| Conv. 1 | $4 \times 4 \times 32$ | 2 | $\times$ |
| Conv. 2 | $4 \times 4 \times 64$ | 2 | $\checkmark$ |
| Conv. 3 | $4 \times 4 \times 128$ | 2 | $\checkmark$ |
| Conv. 4 | $4 \times 4 \times 256$ | 2 | $\checkmark$ |
| Conv. 4 | $4 \times 4 \times 512$ | 2 | $\checkmark$ |
| Conv. 6 | $4 \times 4 \times 768$ | 2 | $\checkmark$ |

(b) Decoder

| Layer | Filter Size | Stride | BN |
|---|---|---|---|
| F-Conv. 1 | $1 \times 1 \times 512$ | 2 | $\checkmark$ |
| F-Conv. 2 | $1 \times 1 \times 256$ | 2 | $\checkmark$ |
| F-Conv. 3 | $4 \times 4 \times 128$ | 2 | $\checkmark$ |
| F-Conv. 4 | $4 \times 4 \times 64$ | 2 | $\checkmark$ |
| F-Conv. 5 | $4 \times 4 \times 32$ | 2 | $\checkmark$ |
| F-Conv. 6 | $4 \times 4 \times 3$ | 2 | $\times$ |

Table 9: Details of the discriminator. The discriminator structure remains the same throughout all the experiments in this work.

(a) Discriminator

| Layer | Filter Size | Stride | BN |
|---|---|---|---|
| Conv. 1 | $4 \times 4 \times 64$ | 2 | $\times$ |
| Conv. 2 | $4 \times 4 \times 128$ | 2 | $\checkmark$ |
| Conv. 3 | $4 \times 4 \times 256$ | 1 | $\checkmark$ |
| Conv. 4 | $4 \times 4 \times 1$ | 1 | $\times$ |

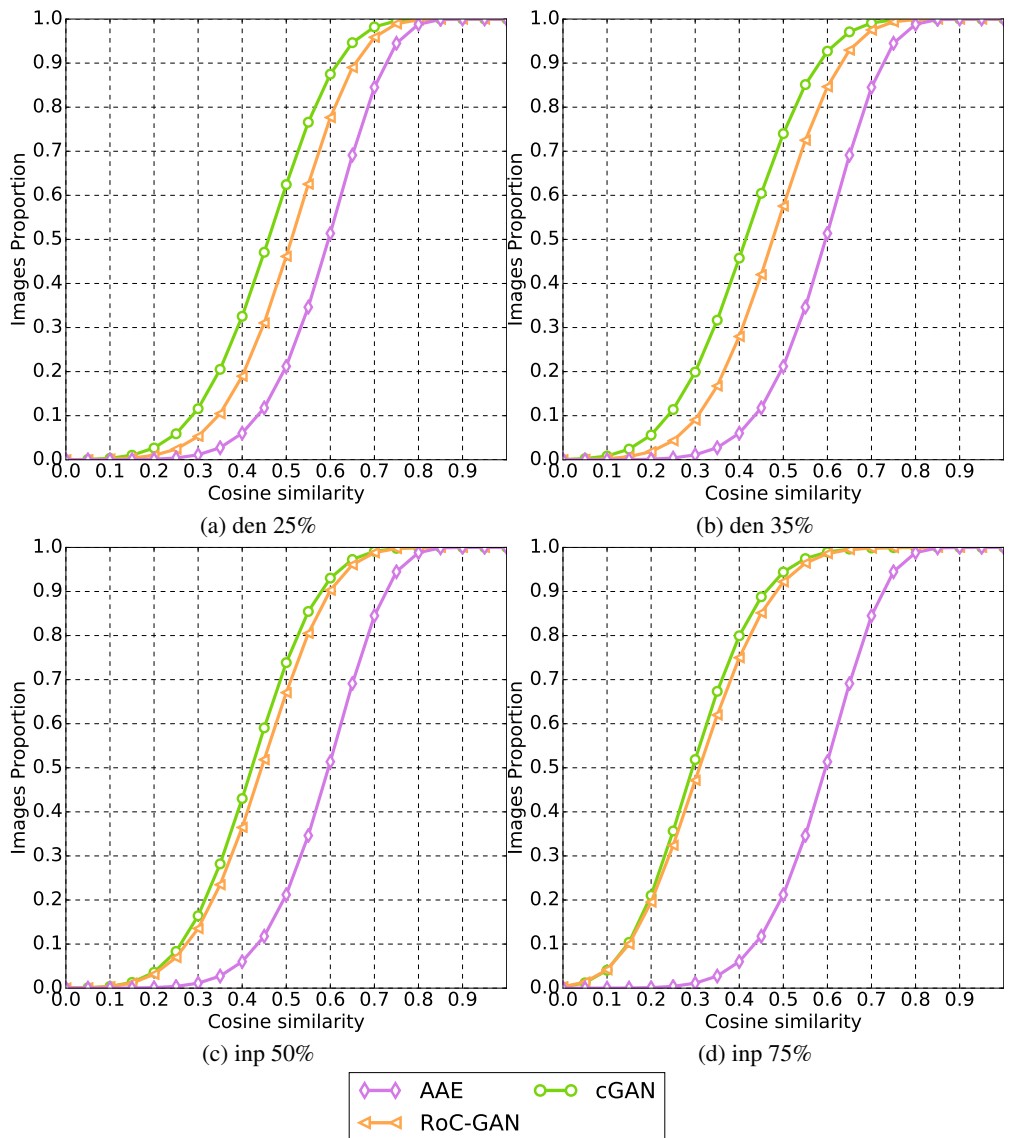

Figure 11: Cosine distance distribution plot (sec. D.4). A perfect reconstruction per compared image would yield a plot of a Dirac delta around one; a narrow distribution centered at one denotes proximity to the target images' embeddings. The word 'den' abbreviates denoising and 'inp' sparse inpainting.

### E.1 HYPER-PARAMETER RANGE

Our model introduces three new loss terms, i.e. $\mathcal{L}_{lat}$, $\mathcal{L}_{AE}$ and $\mathcal{L}_{decov}$ (in the case with skip) with respect to the baseline cGAN. Understandably, those introduce three new hyper-parameters, which need to be validated. The validation and selection of the hyper-parameters was done in a withheld set of images. In the following paragraphs, we design an experiment where we scrutinize one hyper-parameter every time, while we keep the rest in their selected value. During our experimentation, we observed that the optimal values of these three hyper-parameters might differ per case/network, however in this manuscript *the hyper-parameters remain the same throughout our experimentation.*

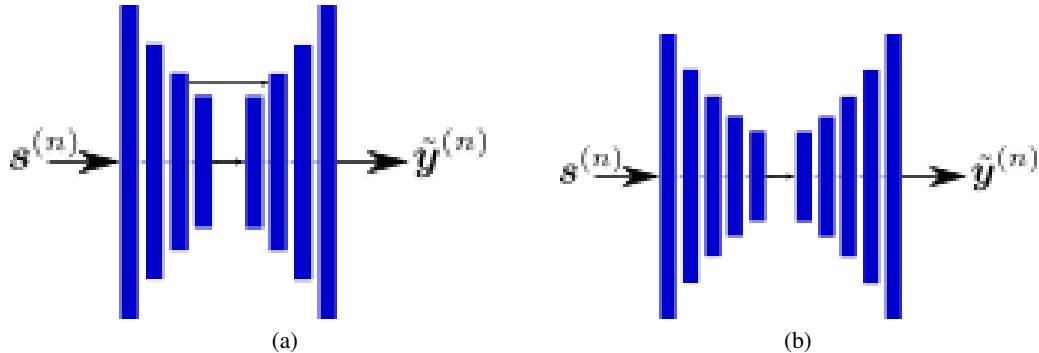

(a)           (b)

Figure 12: The layer schematics of the generators in case of (a) the '4layer-skip' case, (b) the '5layer' case.

The search space for each term is decided from its theoretical properties and our intuition. For instance, the $\lambda_{ae}$ would have a value similar to the $\lambda_c$[7]. In a similar manner, the latent loss encourages the two streams' latent representations to be similar, however the final evaluation is performed in the pixel space, hence we assume that a value smaller than $\lambda_c$ is appropriate.

In table 10, we assess different values for the $\lambda_l$. The results demonstrate that values larger than 10 the results are similar, which dictates that our model resilient to the precise selection of the latent loss hyper-parameter.

| $\lambda_l$ | 1 | 5 | 10 | 15 | 20 | 25 | 30 | 50 | 100 |
|---|---|---|---|---|---|---|---|---|---|
| 25% noise | 0.7936 | 0.8046 | 0.8179 | 0.8234 | 0.8320 | 0.8343 | 0.8312 | 0.8299 | 0.8321 |
| 35% noise | 0.7718 | 0.7820 | 0.7927 | 0.8016 | 0.8150 | 0.8211 | 0.8130 | 0.8138 | 0.8226 |

Table 10: Validation of $\lambda_l$ values (hyper-parameter choices) in the '4layer' network. Unless explicitly mentioned otherwise, all the quantitative results measure the SSIM value. We notice that for $\lambda_l$ larger than 10 the results are significantly better than the baseline.

| $\lambda_{ae}$ | 1 | 10 | 20 | 50 | 100 | 200 | 250 | 300 |
|---|---|---|---|---|---|---|---|---|
| 25% noise | 0.7962 | 0.8193 | 0.8277 | 0.8297 | 0.8343 | 0.8301 | 0.8363 | 0.8238 |
| 35% noise | 0.7765 | 0.8050 | 0.8147 | 0.8158 | 0.8211 | 0.8081 | 0.8218 | 0.8118 |

Table 11: Validation of $\lambda_{ae}$ values (hyper-parameter choices) in the '4layer' network. The network remains robust for a wide range of values of the hyper-parameter $\lambda_{ae}$; for $\lambda_{ae} >= 20$ the test results demonstrate a similar performance.

Different values of $\lambda_{ae}$ are considered in table 11. RocGAN are robust to a wide range of values and both the visual and the quantitative results remain similar. Even though the best results are obtained with $\lambda_{ae} = 250$, we select $\lambda_{ae} = 100$ for our experiments. The difference for the two choices is marginal, thus we choose the value 100 since resonates with our intuition ($\lambda_{ae} = \lambda_c$).

The third term of $\lambda_{decov}$ is scrutinized in the table 12. In our experimentation, the $\lambda_{decov}$ has a different effect per experiment; based on the results of our validation we choose $\lambda_{decov} = 1$ for our experiments.

In conclusion, the additional hyper-parameters introduced by our model can accept a range of values without affecting significantly the results.

---

[7]To fairly compare with baseline cGAN, we use the default value for the loss as mentioned by Isola et al. (2017).

| $\lambda_{decov}$ | 1 | 5 | 10 | 15 |
|---|---|---|---|---|
| 25% noise | 0.8963 | 0.8902 | 0.8868 | 0.8891 |
| 35% noise | 0.8806 | 0.8730 | 0.8693 | 0.8712 |

Table 12: Validation of $\lambda_{decov}$ values (hyper-parameter choices) in the '4layer-skip' network. The network is more sensitive to the value of the $\lambda_{decov}$ than the $\lambda_l$ and $\lambda_{ae}$.

## E.2 SIGNIFICANCE OF LOSS TERMS

To study further the significance of the four loss terms, we experiment with setting $\lambda_* = 0$ alternatingly. Apart from the '4layer' network, we implement the '4layer-skip' to assess the $\lambda_{decov} = 0$ case. The '4layer-skip' includes the same layers as the '4layer', however it includes a lateral connection from the encoder to the decoder.

The experimental results in table 13 confirm our prior intuition that the latent loss ($\mathcal{L}_{lat}$) is the most crucial for our model in the no-skip case, but not as significant in the skip case. In the skip case, the reconstruction losses in both pathways are significant.

| **network** | $\lambda_c = 0$ | $\lambda_{ae} = 0$ | $\lambda_l = 0$ | $\lambda_\pi = 0$ | $\lambda_{decov} = 0$ | **full** |
|---|---|---|---|---|---|---|
| *no-skip* | 0.827 | 0.824 | 0.787 | 0.830 | - | **0.834** |
| *skip* | 0.841 | 0.855 | 0.890 | 0.872 | 0.889 | **0.892** |

Table 13: Quantitative results (SSIM) for setting $\lambda_* = 0$ alternatingly (sec. E.2). In each column, we set the respective hyper-parameter to zero while keeping the rest fixed.

## E.3 ADDITIONAL NOISE

To evaluate whether RoCGAN/cGAN are resilient to noise, we experiment with additional noise. We include a baseline cGAN to the comparison to study whether their performance changes similarly. Both networks are trained with the 25% noise (denosing task).

We evaluate the performance in two cases: i) additional noise of the same type, ii) additional noise of different type. For this experiment, we abbreviate noise as $x/y$ where $x$ depicts the amount of noise in denoising task (i.e. $x\%$ of the pixels in each channel are dropped with a uniform probability) and $y$ the sparse inpainting task (i.e. $y\%$ black pixels). In both cases, we evaluate the performance by incrementally increasing the amount of noise. Specifically, both networks are tested in 25/0, 35/0, 50/0 for noise of the same type and 25/10, 25/20 and 25/25 for different type of noise. We note the networks have not been trained on any of the testing noises other than the 25/0 case.

| *Method/Noise* | 25/0 | 35/0 | 50/0 | 25/10 | 25/20 | 25/25 |
|---|---|---|---|---|---|---|
| Baseline-4layer | 0.8030 | 0.7650 | 0.6642 | 0.7598 | 0.6698 | 0.6114 |
| Ours-4layer | 0.8343 | 0.8211 | 0.7678 | 0.8164 | 0.7649 | 0.721 |
| Diff. | | 3.898% | 7.334% | 15.598% | 7.449% | 14.198% | 17.926% |

Table 14: Quantitative results for the additional noise experiment (sec. E.3). We test the baseline and our model in additional noise cases; as the noise is incrementally augmented the difference is increasing with our model much more resilient to additional noise. This is more pronounced in the different type of noise, i.e. the 25/10, 25/20 and 25/25 cases.

The quantitative results are exported in table 14. The results dictate that while cGAN are volatile to additional amount of noise, RoCGAN's performance is more stable. This becomes even more distinct in the case of different type of noise, where the relative improvement can be up to **17.9%**.

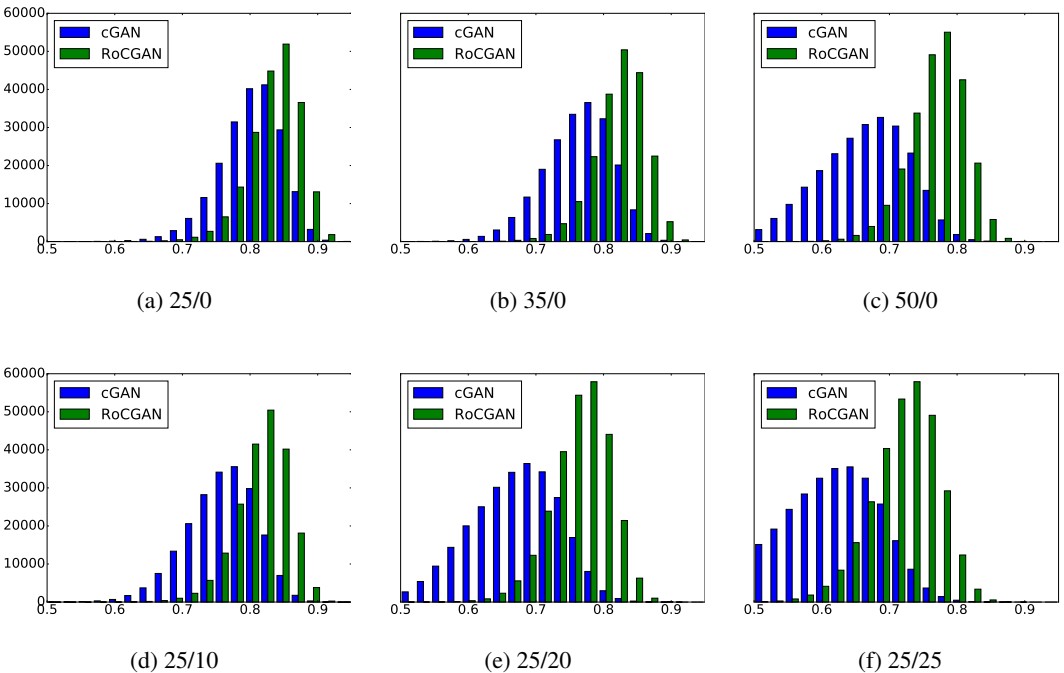

Figure 13: Histogram plots for different noise cases (experiment of sec. E.3). The distribution that is more concentrated to the right of the histogram is closer to the target distribution. In (a) we note that even in the training noise case, the two histograms differ with ours concentrating towards the right. However, as there is additional noise added in (b), (c) the histogram of cGAN deteriorates faster. A similar phenomenon is observed in the cases of (d), (e) and (f).

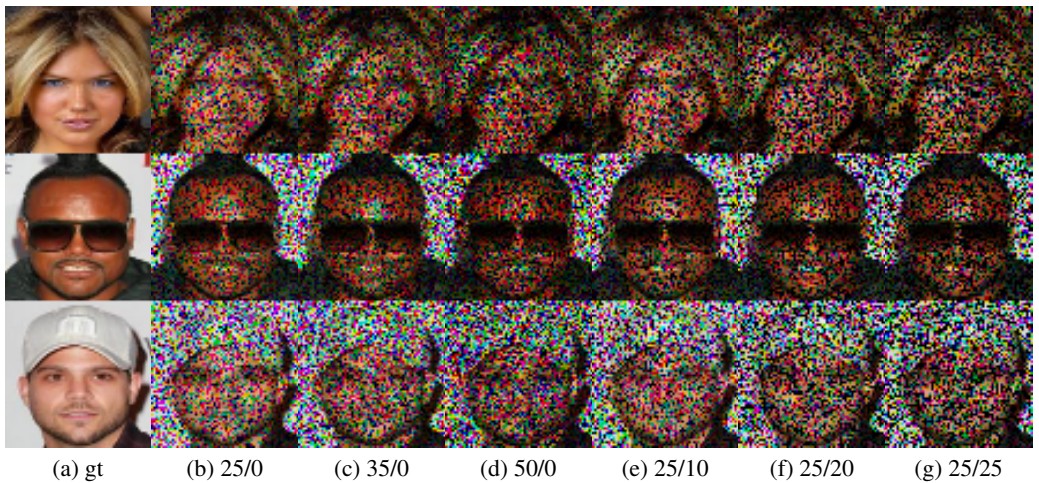

Figure 14: Sample images depicting the corruption level in each case (sec. E.3).

To illustrate the difference of performance between the two models, we accumulate the SSIM values of each case and divide them in 20 bins[8]. In Fig. 13, the histograms of each case are plotted. We note that RoCGAN is much more resilient to increased or even unseen noise. Qualitative results of the

---

[8]In our case the SSIM values lie in the interval $[0.5, 0.95]$; a high SSIM value denotes an image close to the target image.

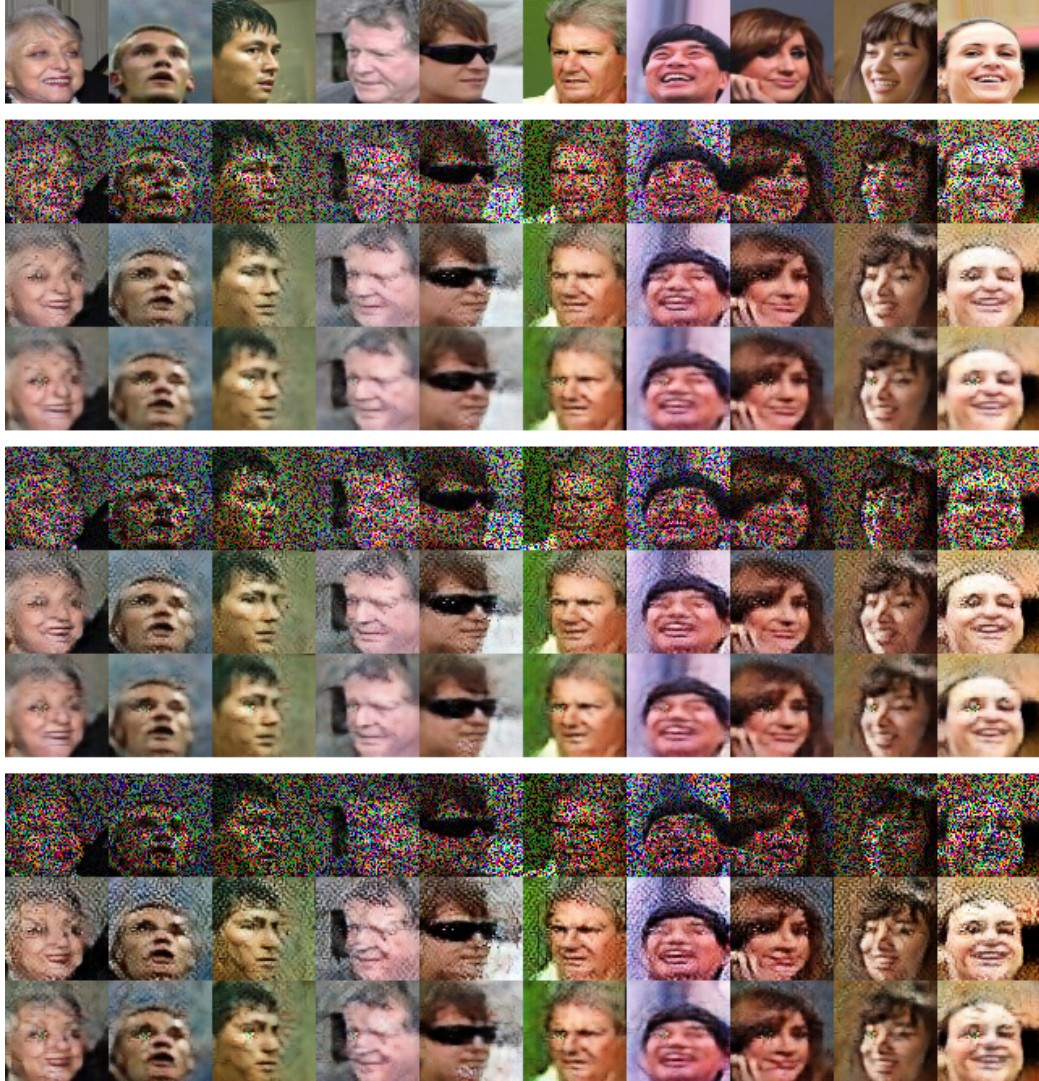

Figure 15: Qualitative figure illustrating the different noise levels (sec. E.3). The first row depicts different target samples, while every three-row block, depicts the corrupted image, the baseline output and our output. The blocks top down correspond to the 25%, 35%, 50% noise (25/0, 35/0 and 50/0). The images in the first blocks are closer to the respective target images; as we increase the noise the baseline results deteriorate faster than RoCGAN outputs. The readers can zoom-in to further notice the difference in the quality of the outputs.

difference are offered in Fig. 15, 16. We consider that this improvement in the robustness in the face of additional noise is in its own a considerable improvement to the original cGAN.

## E.4  ADVERSARIAL EXAMPLES

Apart from testing in the face of additional noise, we explore the adversarial attacks. Recent works (Szegedy et al., 2014; Yuan et al., 2017; Samangouei et al., 2018) explore the robustness of (deep) classifiers. Adversarial attacks modify the input image (of the network) so that the network misclassifies the image. To the authors' knowledge, there has not been much investigation of adversarial attacks in the context of image-to-image translation or any other regression task. However, if we consider adversarial examples as a perturbation of the original input, we explore whether this has any effect in the methods.

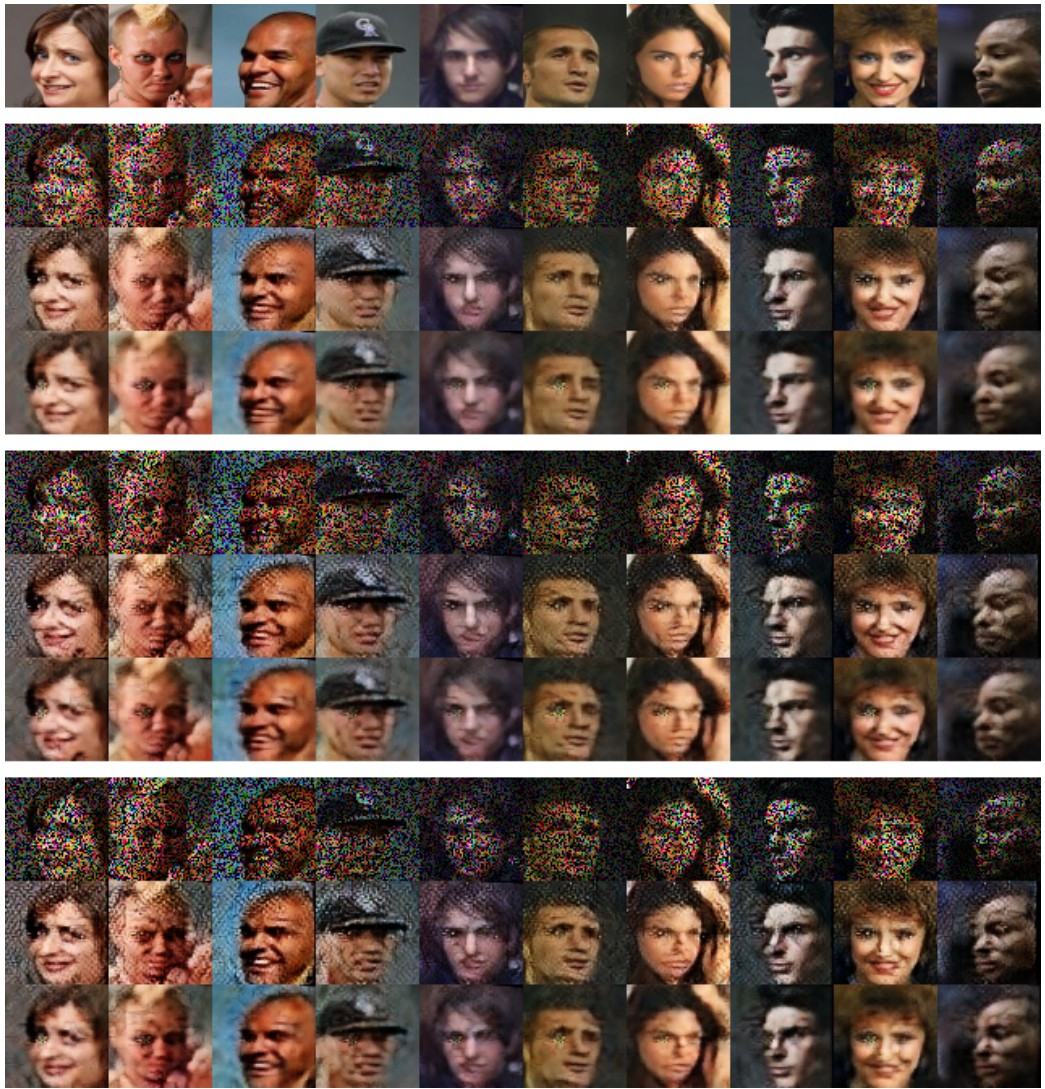

Figure 16: Qualitative figure for different type of noise during testing (see Fig 15). The first row depicts different target samples, while every three-row block, depicts the corrupted image, the baseline output and our output. The blocks top down correspond to the 25/10, 25/20, 25/25 cases (different type of testing noise). The last block contains the most challenging noise in this work, i.e. both increased noise and of different type than the training noise. Nevertheless, our model generates a more realistic image in comparison to the baseline.

Neither cGAN nor RoCGAN are designed to be robust in adversarial examples, however in this section we explore how adversarial examples can affect them. We consider the FGSM method of Goodfellow et al. (2015) as one of the first and simplest methods for generating adversarial examples. In our case, we modify each source signal $s$ as:

$$\tilde{s} = s + \eta \tag{19}$$

where $\eta$ is the perturbation. That is defined as:

$$\eta = \epsilon \, \text{sign}\left(\nabla_s \mathcal{L}(s, y)\right) \tag{20}$$

with $\epsilon$ a hyper-parameter, $y$ the target signal and $\mathcal{L}$ the loss.

In our case, we select the $\ell_1$ loss as the loss between the target and the generated images. We set $\epsilon = 0.01$ following the original paper. The evaluation is added in table 15. The results demonstrate that the baseline is affected more from the added noise, while our model is more robust to it.

| Method/Noise | 25% | 25% + adversarial noise |
|---|---|---|
| Baseline-4layer | 0.8030 | 0.7561 |
| Ours-4layer | 0.8343 | 0.8046 |
| Diff. | 3.898% | 6.414% |

Table 15: Quantitative results for the adversarial examples (sec. E.4). The first column corresponds to the '4layer' case and are added for comparison.

