# OpenReview forum: "Robust Conditional Generative Adversarial Networks"
_ICLR.cc/2019/Conference_

### Official Review · AnonReviewer2 · 2018-11-02
**Explanation is not clear and experiments are weak**

**Rating:** 6
**Confidence:** 4

**Review:**

Authors propose to augment a conditional GAN model with an unsupervised branch for spanning target manifold and show better performance than the conditional GAN in natural scene generation and face generation.

However the novelty is limited and not well explained.
1.Similar idea of using an autoencoder as another branch to help image generation has been proposed in Ma et al.’s work.
Liqian Ma, Qianru Sun, Stamatios Georgoulis, Luc Van Gool, Bernt Schiele, Mario Fritz. Disentangled Person Image Generation, CVPR 2018.

2. In the paper authors claim that skip connection makes it harder to train the longer path, which is kind of contradictory to what is commonly done in tasks of image classification, semantic segmentation and depth estimation. Can authors explain this claim?
In addition, it is not clear why maximizing the variance can address the challenge of training longer path.

3. In Table 1, the improvement over baselines is small in case of sparse inpaint setting.

4. In Figure 4, the fourth row is more blurry than the third row although with less artifacts like black dots.


%%%%%%%% After rebuttal %%%%%%%%

I appreciate authors' efforts to address my comments and am satisfied with their response. I will change decision from rejection to acceptance.

---

> ### Author Response · Authors · 2018-11-16
> **Response to reviewer 2 - part 2 of 2**
>
> 2b) 'In addition, it is not clear why maximizing the variance can address the challenge of training longer path.':
>
> Reducing the correlations of the weights has several advantages that are well-studied in both computer vision and machine learning. Several methods for decorrelating the weights has been used in deep networks, for instance [6-10].
>
> An intuitive idea about why we have included this loss: By reducing the correlations we encourage our method to explore different 'principal' directions in different layers, which is beneficial for training the network. Similar observations and experiments for the benefit of exploring different directions during training have been explored in [4-6].
>
>
> 3) 'Covariance is computed for decov loss but it is not clear which layer’s representation is used to compute covariance.':
>
> We have actually included the requested information in the original submission (sec. 4, page 7 in the revised manuscript).
>
>
> 4) 'In Table 1, the improvement over baselines is small in case of sparse inpainting setting.':
>
> We appreciate the comment; we have performed a similar analysis in sec D.2 (appendix). In short: the improvement in the additional noise experiments (sparse inpainting task) is not marginal but quite significant (up to 15%). Furthermore, we believe the experiments that we have conducted cover several cases and the results are always consistent, i.e. RoC-GAN *always* improve the baseline, while in the regions of more extreme noise or adversarial perturbations the difference is substantial.
>
>
> 5) 'In Figure 4, the fourth row is more blurry than the third row although with less artifacts like black dots.':
>
> We argue that the black dots the reviewer mentions make the images unrealistic. Such irregularities can have detrimental effect for higher level tasks accepting those images as input.  Nevertheless, to demonstrate with quantitative metrics the difference, we have added a new metric for the experiments of faces. The metric is focused on the similarity of the identities of the facial images. We have measured the distance between the methods' outputs and the target images and prepared the cumulative plot. Please find the complete metric analysis in section D.4 in the appendix.
>
>
> [4] Jia, Kui et al. "Improving training of deep neural networks via singular value bounding", CVPR 2017.
> [5] Miyato, Takeru et al. "Spectral normalization for generative adversarial networks", ICLR 2018.
> [6] Bansal, Nitin et al. "Can We Gain More from Orthogonality Regularizations in Training Deep CNNs?", Arxiv.
> [7] Cohen, Taco and Welling, Max "Group equivariant convolutional networks", ICML 2016.
> [8] Cogswell, Michael, et al. "Reducing overfitting in deep networks by decorrelating representations.", ICLR 2016.
> [9] Rodriguez, Pau et al. "Regularizing cnns with locally constrained decorrelations", Arxiv.
> [10] Ozay, Mete and Okatani, Takayuki "Optimization on Submanifolds of Convolution Kernels in CNNs", Arxiv.

---

> > ### Author Response · Authors · 2018-12-05
> > **Thanks for your reply**
> >
> > Dear reviewer,
> >
> > We are thankful to the reviewer for their vote for acceptance after our revision.
> >
> > We are happy to answer any further question, to address any potential remaining concern. We thank the reviewer for the feedback which has helped improve the quality of our manuscript.

---

> ### Author Response · Authors · 2018-11-16
> **Response to reviewer 2 - part 1 of 2**
>
> We appreciate the constructive feedback of the reviewer; in addition to the general comments on top that summarize our revisions, we answer to the reviewer's points below:
>
> 1) 'The novelty is limited and not well explained.':
>
> Even though we might not have emphasized the novelty enough, we believe that our paper makes several contributions. We have pointed them out in the revised text (please consult the general comment above).
> The essence of our contributions is studying robustness of cGAN in dense regression tasks. To the authors' knowledge this has not been studied at all for cGAN, i.e. a widely used framework for dense regression. If the reviewer has noticed it *anywhere*, we are happy to reconsider the positioning of our manuscript.
>
>
> 1) 'Similar idea of using an autoencoder as another branch to help image generation has been proposed in Ma et al.’s work ([1])':
>
> We disagree with the reviewer; the two works differ significantly in their use of AE:
>    a) Ma et al. utilize an autoencoder (AE) with a different goal than RoC-GAN. They use three specialized AE to obtain the latent representations (embeddings) and not to leverage structure in the target (e.g. image) space.
>    b) They devise a well-thought and heavily engineered pipeline for the task of person image generation (Fig. 2, 3 of their paper). Several of their modules, e.g. Region of interest boxes, are task-specific. Our goal is the extension of *any* cGAN to a more robust model.
>    c) The AE in [1] is learned separately (and then is fixed), while the AE pathway in our work is not fixed, but *jointly* optimized with the regression pathway.
>    d) Their loss functions are different. In particular, in [1] the authors include different losses in each step of their two-stage pipeline, while ours is a generic loss that can differ per task.
> In that sense the two works are orthogonal by the use of AE. Despite the differences we consider that some ideas can be used to extend RoC-GAN, e.g. using a discriminator to match the latent representations (as done for the embeddings in [1]). We have added this as future work in the manuscript.
>
>
> 2a) 'In the paper authors claim that skip connection makes it harder to train the longer path, which is kind of contradictory to what is commonly done in tasks of image classification, semantic segmentation and depth estimation. Can authors explain this claim?':
>
> To the best of our knowledge, there has not been much study of how the longer path is optimized in cGAN setting. However, in the broader community of deep learning, several papers report the issue with the longer path training, please check [2-3]. As widely reported, the skip connections might help convergence, but they also enable the network to trivially copy the representations of previous layers and might shatter the meaningful representation learning in the longer path.
>
> A more intuitive explanation in our case: a trivial solution for the network would be to copy the representation to the decoder. The option of copying the meaningful representations through the shortcut is more 'attractive' in our case due to the latent loss Llat. If we do not regularize the longer path representations, the network has less incentive to learn meaningful representations in the longer path, which defeats the concept of including the latent loss. On the contrary, including by decorrelating the weights we encourage both pathways to learn meaningful representations in the longer path.
>
> [1] Ma, Liqian, et al. "Disentangled Person Image Generation", CVPR 2018.
> [2] Rasmus, Antti, et al. "Semi-supervised learning with ladder networks.", NIPS 2015.
> [3] Zhang, Yuting, et al. "Augmenting supervised neural networks with unsupervised objectives for large-scale image classification", ICML 2016.

---

### Official Review · AnonReviewer3 · 2018-11-02
**Well-written paper but the novelty and significance might be a weakness**

**Rating:** 6
**Confidence:** 4

**Review:**

General:
In general, this is a well-written paper. This work focuses on the robustness of conditional GAN(RoC-GAN) when facing the noise. The authors claim the generator of RoC-Gan will span the target manifold, even in the presence of large amounts of noise. The main contribution of the paper is to introduce a two-pathway model, where one of them is used to perform regression as ordinary GAN while the other one helps the whole model span the target domain.

Strength:
1. The idea is simple and straightforward. The authors provide necessary theoretical analysis and empirical validation for their model.
2. The proposed method seems technically correct to me. i.e. Although I am not very sure how well it works in practice, the idea is fine.

Possible Improvements:
1. I agree adding another auto-encoder as a helper may give better generation results by spanning the whole target space, but I don't think this constraint is strong enough in practice.
2. In section 3.3, the time complexity of computing 'L_deconv' seems extremely large. From the perspective of numerical optimization, optimizing such a matrix will cause trouble if the dimension of weight matrices are large. i.e. optimizing the high-dimensional covariance matrix seems a problem to me.
3. The experiments looks good. The experiments could be more convincing if using more complex data sets(e.g. CIFAR10, ImageNet) besides CelebA. My concern for using such data sets(the resolution of images is low and the distribution is simple)  is that: although the noise seems to corrupt most of the image, the distribution of the image is not complex, so the generative model can recover it easily. Since this is a more empirical paper, the experiments should be more convincing.

Conclusion:
The author(s) are thoughtful and they put lots of work on this paper. The proposed method is simple. For novelty and significance, I think the idea is not very fancy to me. I am not very convinced by the method proposed in the paper. Although the paper demonstrates the robustness of their model with different experiments, most of them were not performed on deep neural networks and complicated data sets. As a conclusion, I vote for weak rejection.

Minor suggestion:
Increase the resolution of the figures.

------------------------------- After Rebuttal ---------------------------------
I am very satisfied with the authors' response, so I will change my vote from rejection to acceptance.

---

> ### Author Response · Authors · 2018-11-12
> **Response to reviewer 3**
>
> We thank the reviewer for recognizing the effort and contribution of our method. In addition to the general answer above, we answer each of the improvement points below:
>
> 1) 'I agree adding another auto-encoder as a helper may give better generation results by spanning the whole target space, but I don't think this constraint is strong enough in practice. ':
>
> We respectfully disagree with the reviewer; we have demonstrated in a series of experiments how this modification is beneficial. We provide the intuition, the synthetic experiment, a linear analogy analysis, and several experiments.
> In the revision we add a visual example for the linear subspace (appendix, sec. B). We demonstrate how one corrupted and one clean image can have similar reconstructions from a PCA model.
>
>
> 2) 'In section 3.3, the time complexity of computing '$L_deconv$' seems extremely large. From the perspective of numerical optimization, optimizing such a matrix will cause trouble if the dimension of weight matrices are large. i.e. optimizing the high-dimensional covariance matrix seems a problem to me.':
>
> In the implementation details (section 4; page 7), we mention that we use $L_deconv$ in the output of the encoders. Those layers include tensors of $batch x 1 x 1 x channels$ where the number of channels is typically up to 1024. In our experiments for 4 layer network channels=512, for the 5 and 6 layer networks channels=768. Cogswell et al. ([1]) include an analysis for deeper networks. In practice, we have not noticed a significant computational burden, but this can be further explored in the future.
>
>
> 3) 'The experiments look good. The experiments could be more convincing if using more complex data sets(e.g. CIFAR10, ImageNet) besides CelebA. My concern for using such data sets(the resolution of images is low and the distribution is simple)  is that: although the noise seems to corrupt most of the image, the distribution of the image is not complex, so the generative model can recover it easily. Since this is a more empirical paper, the experiments should be more convincing.':
>
> We have conducted the requested experiments on Imagenet (appendix, sec. D.1). The results are in line with those reported in the main paper -- particularly the natural scenes case and confirm the advantages of our method. That is, RoC-GAN outperform the cGAN in both denoising and sparse inpainting while the difference is increased when evaluated with additional noise.  We note that the same hyper-parameters (as the rest of the paper) are used; additional tuning per experiment might be beneficial however for avoiding confusion all the hyper-parameters remain the same.
>
>
> 4) 'Although the paper demonstrates the robustness of their model with different experiments, most of them were not performed on deep neural networks and complicated data sets.':
>
> In the revised version we have included an experiment on Imagenet (appendix, sec. D.1). Our model is not architecture dependent. Specifically, RoC-GAN can be seen as a meta algorithm which can be used to augment any existing cGAN model to achieve additional robustness. We have made our best effort to demonstrate that with i) similar types of noise, ii) types of noise 'unseen' during training, iii) adversarial perturbations.
>
> 5) 'Minor suggestion: Increase the resolution of the figures.':
>
> We will add new figures to reflect the AAE method added and improve older ones the next few days. We appreciate the proposal.
>
> [1] Cogswell, Michael, et al. "Reducing overfitting in deep networks by decorrelating representations.", ICLR 2016.

---

> > ### Author Response · Authors · 2018-12-02
> > **Post-rebuttal concerns?**
> >
> > Hello,
> >
> > thanks once again for the time and effort to review our manuscript. Given that the rebuttal period is closing soon, we would appreciate any additional feedback and re-assessment of our submission.
> >
> > The main concerns of the reviewer were twofold: i) experimenting on a more established dataset and ii) the novelty. To address the former, we have included experiments on Imagenet. The same outcomes as with the rest of the experiments are observed, i.e. RoC-GAN model outperforms cGAN counterpart in all cases. Regarding the latter (i.e. novelty), we have included bullet-point contributions in the revised text. In addition, we have written a detailed report in the general comments above elaborating on the novelty.
> >
> > In addition to those two improvements, we have also included several other experimental improvements. For instance, we include a new comparison method as upper bound of both networks and new evaluation metrics (please check the general comments for details). Those changes have made our manuscript even stronger; as the reviewer recognized we have made an effort to include several experiments already.
> >
> > We believe that the comments of the reviewer have made the manuscript stronger, therefore we would welcome any additional feedback. Especially, if the reviewer has any new concerns that have not been addressed before.

---

### Official Review · AnonReviewer1 · 2018-11-05
**Simple but effective method, methodological novelties are limited though!**

**Rating:** 6
**Confidence:** 4

**Review:**

This manuscript proposes a robust version of conditional GAN (named RoC-GAN) that leverage the intrinsic structure in the output space. To achieve robustness, the authors replace the single pathway in the generator with two different pathways that partially share weights. The authors study the theoretical properties of RoC-GAN and prove that it shares the same properties as the vanilla GAN. For quantitative evaluations, the authors use two datasets of natural scenes and faces and evaluate denoising and sparse inpainting using the SSIM metric.
-	The idea is simple and seems to be working. The methodological novelties seem more-or-less limited, but the theoretical analysis and the intuitive (and well-motivated) modification over CGANs add merits to the paper.
-	The theoretical analysis of the method relates RoC-GAN to the original GAN, rather than CGAN! What is the connection here? If RoC-GAN is very similar to CGAN from a theoretical point of view (which it seems to be), then all the analysis to relate it to traditional GAN seem useless.
-	The extensive experiments in the supplementary material are appreciated. But the authors only compare their method with one single previous work (i.e., Rick Chang et al. (2017)), while there are several similar related works (either based on adversarial training strategies or simple denoising AEs).
-	Also, ablation studies can further show how each component of the model contributes to the final results. What if we were to only use the two-path generator without adversarial training? Different components of the final loss function can be removed and analyzed one at a time!
-	What are the conditions for mode-collapse for the proposed GAN? There are no discussions on this.

---

> ### Author Response · Authors · 2018-11-12
> **Response to reviewer 1**
>
> In addition to the general answer above, we answer each of the points raised below:
>
> 1) 'The idea is simple and seems to be working. The methodological novelties seem more-or-less limited, but the theoretical analysis and the intuitive (and well-motivated) modification over CGANs add merits to the paper. ':
>
> We thank the reviewer for the recognition of the work.
>
>
> 2) 'The theoretical analysis of the method relates RoC-GAN to the original GAN, rather than CGAN! What is the connection here? If RoC-GAN is very similar to CGAN from a theoretical point of view (which it seems to be), then all the analysis to relate it to traditional GAN seem useless.':
>
> On the contrary, we prove that our RoC-GAN shares the same theoretical properties as GAN; this can be seen as a sanity check, conforming that our method shares some beneficial theoretical properties with well-studied methods.
> Similar proofs are provided in other extensions to cGAN, such as Zhe et al. ([1]).
>
>
> 3) 'The extensive experiments in the supplementary material are appreciated. But the authors only compare their method with one single previous work (i.e., Rick Chang et al. (2017)), while there are several similar related works (either based on adversarial training strategies or simple denoising AEs).':
>
> The goal of this work is not to propose a state-of-the-art network per se, but rather to present a method that is more robust to additional sources of noise. Our model is not architecture dependent. Specifically, RoC-GAN can be seen as a meta algorithm which can be used to augment any existing cGAN model to achieve additional robustness.
> We scrutinize the robustness under:
> i) similar types of noise,
> ii) types of noise not encountered during training,
> iii) adversarial perturbations.
> In addition to those, we also note that our method performs favorably when tested with samples similar as the training distribution. We have included an external  method to illustrate that even strong performing networks can have difficulty in such tasks.
> However, if there are some specific works that the reviewers feel are particularly relevant, we are happy to evaluate their pre-trained models.
>
>
> 4) 'Ablation studies can further show how each component of the model contributes to the final results. What if we were to only use the two-path generator without adversarial training? Different components of the final loss function can be removed and analyzed one at a time!':
>
> We appreciate the reviewer's proposal; indeed in the synthetic experiment we optimize only the generators to simplify the problem; please see sec. 3.4. In addition, removing one by one the losses is performed in sec. E.2 (appendix).
> Even though our experiments are not exhaustive, we consider that we have covered a wide range of choices; those demonstrate the merits or trade-offs of our RoC-GAN.
>
>
> 5) 'What are the conditions for mode-collapse for the proposed GAN? There are no discussions on this.':
>
> We follow the same strategy as popular methods in cGAN ([2], [3]). We agree with the reviewer that mode collapse is significant especially in original GAN training, however there are other works tackling this issue, e.g. [4-6].
>
>
> [1] Gan, Zhe, et al. "Triangle generative adversarial networks.", NIPS 2017.
> [2] Isola, Phillip et al. "Image-to-Image Translation with Conditional Adversarial Networks", CVPR 2017.
> [3] Zhu, Jun-Yan et al. "Toward multimodal image-to-image translation", NIPS 2017.
> [4] Che, Tong et al. "Mode Regularized Generative Adversarial Networks", ICLR 2017.
> [5] Anonymous, "Generative Adversarial Network Training is a Continual Learning Problem", under review ICLR 2019.
> [6] Anonymous, "DISTRIBUTIONAL CONCAVITY REGULARIZATION FOR GANS", under review ICLR 2019.

---

> > ### Comment · AnonReviewer1 · 2018-11-21
> > **Response to the authors' rebuttal**
> >
> > Thanks for the response. The points raised in my previous review are responded accordingly. I stand with my previous assessment and decision on the paper.

---

> > > ### Author Response · Authors · 2018-12-05
> > > **Thanks - any additional questions?**
> > >
> > > Dear reviewer,
> > >
> > > thanks once again for the effort to review our manuscript and reply to our revision. We strongly believe that the revisions have improved our manuscript.
> > >
> > > Based on the reply, we hold that we addressed the comments the reviewer raised. If not, it would be beneficial for us to understand any further concerns.
> > >
> > > The main issues the reviewer raised were twofold: the comparisons in the experiments and the contribution of our work. For the former, we trained a method that quantifies the representation power of each network (i.e. upper bounds the compared methods). We addressed the novelty in the revised text explicitly (also elaborated in the general comment). Are there any additional arguments or experiments that would persuade the reviewer for the significance of robustness in dense regression tasks?
> > >
> > > In addition to the aforementioned revisions, we have included additional experiments (imagenet) and a new metric (see general comment). Based on our revision, reviewers 2 and 3 posted a positive response and increased their ratings.
> > >
> > > We would therefore ask the reviewer whether there are any additional concerns that we could address in our manuscript. This is significant, especially given that we have made our effort to conduct all the requested experiments and even extend our manuscript (e.g. devise bounds for our method).

---

### Author Response · Authors · 2018-11-12
**General comment - revision 1**

We appreciate the constructive feedback of the reviewers. In response to their reviews, we have updated several sections in the paper:

* We add an experiment on the ImageNet dataset, in both sparse inpainting and denoising (appendix, sec. D1).
* We include a paragraph with the contributions in the introduction.
* We train a method to demonstrate the  maximum representational power of each network (this can be thought of as the upper bound of each experiment). Specifically, we train an adversarial autoencoder (in the target space) and utilize the reconstructed images for evaluation.
* We include a visual example to illustrate how a projection in a linear subspace can promote the output to span the target subspace (appendix, sec. B).

In summary, we believe that the aforementioned results strengthen our claims and improve the paper.

---

### Author Response · Authors · 2018-11-16
**General comment - revision 2**

One of the crucial issues raised by the reviewers is the novelty of our approach. In this comment we address this general question and also outline the additional revisions (see also general response for the first part):

We argue that there is considerable novelty in our work; to our knowledge there has *not* been a study of robustness in dense regression.

Machine learning is entering an era that is widely used in many diverse applications ranging from particle physics [1] to cyber-security [2] and from medical applications [3] to molecule generation [4]. In several applications and domains safety/robustness is critical, however the majority of the dense regression networks just report the best results ignoring the out of manifold or noise of real applications. Hence, we argue that robustness analysis should be introduced and performed in dense regression tasks.

In this context we restate our contributions:

* We introduce RoC-GAN that performs conditional image generation by leveraging structure in the target space. Neither the model has emerged before, nor the context of our analysis.

* We perform a robustness analysis through a series of experiments. We scrutinize the performance of the original cGAN and our model under different types of noise. We extend the adversarial perturbations in dense regression tasks.

* We experimentally demonstrate how our method can be used with different architectures and tasks. We additionally show that RoC-GAN can be beneficial in semi-supervised learning task or how it performs with lateral connections from encoder to decoder.

To address the request of reviewer 2 for an alternative metric to SSIM, we compute a cumulative plot that measures the similarities of the identities in the face experiment. We employ the well-studied (and robust) recognition embeddings of FaceNet ([5]) to evaluate the similarity of the target image with the outputs of compared methods. For each pair of output and corresponding target image, we compute the cosine distance of their embeddings; the cumulative distribution of those distances is plotted. The plots illustrate that indeed RoC-GAN outputs are closer to the target images with respect to the identities. Please find the complete metric analysis in section D.4 in the appendix.

We consider that our work has become significantly stronger with the revised experimental results, thus we request the reviewers to reconsider their rating.


[1] de Oliveira, Luke et al. "Learning particle physics by example: location-aware generative adversarial networks for physics synthesis", Computing and Software for Big Science 2017.
[2] Ye, Guixin et al. "Yet Another Text Captcha Solver: A Generative Adversarial Network Based Approach", ACM SIGSAC Conference on Computer and Communications Security 2018.
[3] Wei, Wen et al. "Learning Myelin Content in Multiple Sclerosis from Multimodal MRI through Adversarial Training", arxiv 2018.
[4] De Cao, Nicola and Kipf, Thomas "MolGAN: An implicit generative model for small molecular graphs", arxiv 2018.
[5] Schroff, Florian et al. "Facenet: A unified embedding for face recognition and clustering", CVPR 2015.

---

### Author Response · Authors · 2019-03-18
**Open source implementation of RoCGAN**

We have released an open source version of the RoCGAN:
https://github.com/grigorisg9gr/rocgan

This is a Chainer implementation; the network layers and the hyper-parameters might differ from the original used for the ICLR paper, but the relative experimental performance remains the same (as reported in the paper).

---

### Meta-Review · Area_Chair1 · 2018-12-14

**Confidence:** 4
**Recommendation:** Accept (Poster)

**Metareview:**

The proposed method suggests a way to do robust conditional image generation with GANs. The premise is to make the image to image translation model resilient to noise by leveraging structure in the output space, with an unsupervised "pathway".

In general, the qualitative results seem reasonable on a a number of datasets, including those suggested by reviewers. The method appears simple, novel and easy to try.  The main concerns seem to be that the idea is maybe too simple, but I'm not particularly bothered by that. The authors showed it working well on a variety of tasks (synthetic and natural), provide SSIM numbers that look compelling (despite SSIM's short-comings) and otherwise give compelling arguments for the technical soundness of the approach.

Thus, I recommend acceptance.